# A survey on modeling dynamic business processes

Diana Kalibatiene[1] and Olegas Vasilecas[2]

[1] Department of Information Systems, Vilnius Gediminas Technical University, Vilnius, Lithuania
[2] Institute of Applied Computer Science, Vilnius Gediminas Technical University, Vilnius, Lithuania

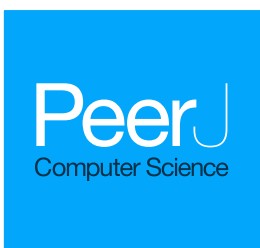

## ABSTRACT

Dynamic and flexible systems offer huge advantages for businesses in addressing dynamic uncertain factors and implementing dynamic business processes (DBP). However, DBP remains a challenge from the perspectives of modeling, simulation, and implementation because of a nontrivial understanding of "*What is a dynamic business process?*" A variety of approaches for DBP modeling and implementation have been proposed over the past years, yet few comprehensive studies analyzing DBP from different particular perspectives (e.g., business process (BP) variability, aspect oriented BP, service compositions, etc.) and research questions that lay the foundation for the development of a meaning of a DBP have been reported. The motivation behind this review is to examine DBP meaning from a global perspective and, consequently, answer the previously presented research question. Therefore, in this paper, we present a systematic literature review (SLR) comprised of 67 papers from five respective digital libraries, which index Computer Science (CS), Information Systems (IS), and Software Engineering (SE) journals and conference proceedings. Two points of view are analyzed in the selected papers. First, we observe the similarities and differences between the proposed approaches to DBP modeling and implementation. From these observations, we define six main requirements for DBP (DBPR). In addition, the comparison of the selected papers according to DBPR shows that most of the approaches analyzed limit BP dynamicity, since they use partially predefined BP models. Secondly, we analyze the papers based on a visualization perspective that shows the less explored areas as follows: more flexible process modeling approach and its implementation in IS should be developed; usage of historic data should be extended; domain knowledge usage, like goal-orientation, multi-criteria optimization, domain knowledge, artificial intelligence, etc., should be included and extended to ensure BP dynamicity. As such, this study makes important contributions and serves as a useful resource for future DBP studies and practice. Moreover, we expect that our results could inspire researchers and practitioners towards further work aimed at bringing forward the field of DBP modeling and implementation.

Corresponding author
Diana Kalibatiene,
diana.kalibatiene@vilniustech.lt

## INTRODUCTION

Business processes (BP) are dynamic due to the changing nature of their environment (*van der Aalst, Pesic & Schonenberg, 2009*; *Pang et al., 2011*; *Vasilecas, Kalibatiene & Lavbič, 2016*). This can include regulatory adaptations (e.g., changes in raw material prices), market evolution (e.g., stock price changes), changes in customer behavior (e.g., rapid change in customer needs), process improvement, enterprise policy shifts, and exceptions defined through business rules (BR). Therefore, as presented in *Pang et al. (2011)*, BPs must be able to support structural and functional changes, such as adding a new activity, substituting an activity, and changing the execution order of multiple activities. Consequently, the models that represent them should be able to reflect this dynamicity. However, classical BP models are intended to represent predefined processes that are more or less structured, and thus do not incorporate enough flexibility. Moreover, dynamic changes usually take place in a run-time environment, i.e., involve making changes "on the fly", and we cannot abort BPs and restart them from the beginning after changes. In this context, BP dynamicity is defined across a spectrum. This ranges from completely static BP (a dynamicity level of zero percent), where all activities and their sequences are predefined in the model without the possibility of changing them, to completely dynamic BP (a dynamicity level of 100%), where each subsequent process activity is chosen from a predefined set of activities, or even defined as required (according to the internal or external context, or historical data) by rules.

Some level of dynamicity is represented by ad-hoc (*Reichert & Dadam, 2009*; *Bizagi, 2021*), agile (*Gong & Janssen, 2012*), flexible (*Rosa et al., 2017*; *Reichert & Weber, 2012*), adaptive (*Reichert & Dadam, 2009*), variable (*Rosa et al., 2017*), customizable (*Rosa et al., 2017*), declarative (*Jimenez-Ramirez, Barba & Del Valle, 2018*; *Eshuis, 2018*), and dynamic BP (*Vasilecas et al., 2016*). However, as outlined in (*Gong & Janssen, 2012*), the concepts of agility and flexibility overlap nevertheless, and in representing a system's ability to respond to environmental changes, agility has a stronger influence than flexibility on the characteristic of speed. According to *Pucher (2010)*, dynamic BP are meant to be a variant of agile processes and enable a business user to make changes in the process at run-time, e.g., by selecting a different sub-process at predefined decision points. *Kiedrowicz (2017)* has noted that dynamics are obtained during process implementation, with an initially non-defined form of such processes completely or in part. In *Vasilecas et al., (2016)*, authors define dynamic BP as a set of activities, which might change at a certain point in time due to the changes occurring in the BP context. Therefore, in dynamic BP a sequence of activities cannot be predefined in advance. According to some (*Rosa et al., 2017*; *Reichert & Weber, 2012*), processes in which customization decisions are made at run-time are known as flexible processes.

According to *Rosa et al. (2017)*, variable BP models families of BP variants, where a family of BP variants is presented via a single model, from which each variant can be derived via certain transformations of the model. Moreover, *Rosa et al. (2017)* have named a consolidated model of process variants by concept customizable process model, from

which the individual variants can be derived via transformations, for example, adding or deleting fragments. Decisions between process variants can be made either at design time or during run-time. A design-time customization affects all instances of the customized process executed in this setting. In contrast, a run-time customization is punctual and affects only one or very few process instances. However, all types of variable processes have predefined process models. A variable BP consists of variable and non-variable segments and only the variable segments of a process could be changed (*Alférez et al., 2014*), at run-time. *Aiello, Bulanov & Groefsema (2010)* remark that variability is very closely related to flexibility. Flexibility offers adaptation and the potential change of a process, whereas variability deals with different versions of a process.

According to some (*Jimenez-Ramirez, Barba & Del Valle, 2018*; *Eshuis, 2018*), declarative models specify what should be done without specifying how it should be done. *Goedertier, Vanthienen & Caron (2015)* state that these models specify a set of constraints, BRs, event conditions, or other (logical) expressions that define the properties of, and dependencies between, activities in a BP. So, BP instances are constructed according to predefined BRs and do not violate them. These rules restrict the final execution path without defining the process model. *Eshuis (2018)* states that declarative artifact-centric process models, which present *knowledge-intensive* processes, use BRs that define how knowledge experts can make progress in a process. However, in many business situations knowledge experts have to deal with uncertainty, which cannot be modelled by BRs in a classical way, assuming clear-cut boundaries between different states of the world. Therefore, *Eshuis (2018)* proposes to model uncertain situations using fuzzy logic and to extend declarative artifact-centric process models with fuzzy sentries.

Declarative BP modeling approaches are based on rules (such as (*Eshuis, 2018*; *Zugal et al., 2015*, etc.). However, as stated in *Goedertier, Vanthienen & Caron (2015)*, the differences between declarative BP modeling approaches lie in a different perception of main elements as the following. Business concerns, including the ConDec language (*Pesic & van der Aalst, 2006*; *Pesic, Schonenberg & van der Aalst, 2007*; *Pesic et al., 2007*), and the PENELOPE language (*Goedertier & Vanthienen, 2006*), only allow the expression of BRs relating to sequence and timing constraints, i.e., the control flow aspects (*Heinl et al., 1999*). The state space describes a discrete set of relevant BP states in terms of the entity types that occur in that space, such as the event history in ConDec and the system time in PENELOPE. Constraint types in ConDec express temporal constraints that must hold between activities in a trace, and PENELOPE discusses business constraint types that are essentially temporal deontic assignments. Knowledge representation and reasoning paradigms can also differ, like Linear Temporal Logic (LTL) in ConDec and the Event Calculus in PENELOPE.

In summarizing the definitions above, some authors use terms such as "adaptive" and "flexible" as synonyms, whilst others hold those concepts as distinct. Therefore, there is still some uncertainty around which approach maps to which level of dynamicity of a BP.

Consequently, one of the objectives of this research is to analyze the level of dynamicity achieved in the proposed approaches, and thereafter the subsequent degree of implementation achieved. So, the main research questions of this study are the following:

RQ1: *What level of dynamicity is achieved by the existing approaches?*

RQ2: *What are the differences between approaches to modeling dynamic BP?*

RQ3: *What are the features of a dynamic BP?*

RQ4: *What are the general limitations or research gaps that exist in the literature on dynamic BP modeling that may require further work?*

RQ5: *What literature reviews on BP dynamicity are known?*

RQ6: *What are the levels of description and refinement of the BP dynamicity approaches analyzed?*

A comprehensive state-of-the-art review is needed to better understand the current state of knowledge in the field of dynamic BP (DBP), and to answer the questions formulated. Although there are a number of scattered studies reviewing different approaches to achieving some level of dynamicity, there is a need to analyze them collectively, and thus to obtain a global view of modeling such BP. Put simply, there is a need to compare levels of dynamicity of BP.

Therefore, this paper draws together a systematic literature review (SLR) of approaches to modeling BP with some level of dynamicity, identifies and classifies main approaches in the field, and provides a comparative evaluation.

## Rationale for the review

Nowadays, BP are dynamic due to the changing nature of their environment, which include regulatory adaptations (e.g., changes in raw material prices), market evolution (e.g., stock price changes), changes in customer behavior (e.g., rapid change in customer needs), process improvement, enterprise policy shifts, and exceptions defined through business rules (BR). Therefore, BPs must be able to support structural and functional changes. However, classical BP models are intended to represent predefined processes that are more or less structured and do not incorporate enough flexibility. Moreover, dynamic changes usually occur in a run-time environment, and we cannot abort BPs and restart them from the beginning after changes. Some dynamicity level is represented by ad-hoc, agile, flexible, adaptive, variable, customizable, declarative, and dynamic BP (DBP). However, there is still some uncertainty around which approach maps to which level of dynamicity of a BP. In the absence of a clear understanding of what is DBP, a comprehensive analytical study is necessary. This SLR provides a clear and comprehensive overview of the available evidence. In addition, our work helps to identify research gaps in our current understanding of the field. The objective of this review is to provide a basis for the developers of DBP models. Our findings laid the foundation for intelligent BP.

## Intended audience

The article is intended to support academic and industry researchers working on DBP. We expect that our results inspire researchers and practitioners for further work aiming at bringing forward DBP modeling and implementation.

The remainder of this paper is organized into a number of sections. "The Scope of the Survey" presents the scope of the survey by defining the main terms and concepts associated with BP and their level of dynamicity. "Approaches to Achieving Dynamicity of BP" presents related works and existing literature reviews on BP and different levels of dynamicity. "The Definition of a Dynamic BP (DBP)" describes the research method used for the analysis of BP with different levels of dynamicity. "Results" presents the results of the analysis performed on the basis of the method defined in the previous section. In "Discussion, Answers to the Research Questions and Future Work", we discuss the results of the analysis and answer the research questions. "Conclusions" then concludes the paper.

## The scope of the survey

A number of approaches propose different ways of modeling and implementing dynamic BP. Their differences lie in their understandings of the concept of dynamic. Therefore, we start our survey with the concepts used to define what it is to be dynamic.

## The definition of the term "dynamic"

In different dictionaries, the term dynamic is understood as *continuously moving or changing* (https://www.ldoceonline.com/dictionary/dynamic) or *constant change or motion* (https://www.yourdictionary.com/dynamic). A dynamic process is therefore one that *constantly changes and progresses* (https://www.collinsdictionary.com/dictionary/english/dynamic) a process of agile change, action, and/or progress (https://www.igi-global.com/dictionary/dynamic-process/69549). According to IGI-Global dictionary (https://www.igi-global.com/dictionary/dynamic-process/69549) and *Reichert & Dadam (2009)*, dynamic process change "refers to a (structural) change that is applied to the schema of a running process instance during run-time. After the change, process execution continues based on the new schema version of the process instance." Ad-hoc process change refers to a process change which is applied in an ad-hoc manner to a given process instance, and is necessary to deal with exceptions or situations not anticipated at the process design stage (*Reichert & Dadam, 2009*). Adaptive process refers to the ability of the process-aware information system (PAIS) to dynamically adapt the schema of ongoing process instances during run-time. As presented in *Reichert & Dadam (2009)*, process schema evolution "refers to the continuous adaptation of the schema of a particular process type to cope with evolving needs and environmental changes." The need for a dynamic change is detected in long-running processes, and then becomes necessary in order to migrate already running process instances to the new schema version (*Reichert & Dadam, 2009*). According to Oracle (https://docs.oracle.com/en/cloud/paas/integration-cloud/user-processes/create-dynamic-process1.html#GUID-99737D0A-DF4C-484E-820A-D30108073951), a dynamic process is a collection of activities or tasks without a predetermined sequence of execution. It provides flexibility for knowledge workers to define the process flow at run-time based on the information available to them. Oracle allows the creation of a dynamic process by selecting a particular pattern or starting from scratch. In addition, a user should define all activities that can be used within a dynamic process. During run-time, knowledge workers decide the course of activities in a

dynamic process, start and complete activates, assign activities to roles, and complete and close process instances. Moreover, each dynamic process has milestones that are sub-goals defined within a process, and are used to track progress.

A BP model can be seen as a scheme defining the "algorithm" of process execution (*van der Aalst, Weske & Grünbauer, 2005*). During the execution of a BP, the system uses the BP model as a "recipe" to determine the sequence of activities to be executed. A BP model is described by a specific BP modeling language, such as BPMN, which plays a determining role in the way that the prescribed process can be executed. The weaker this prescription is, the easier it is to deviate from the predefined process (*Pesic & van der Aalst, 2006*). However, most process models enforce the prescribed procedure without deviations. Flexible BP should allow users to deviate from the prescribed execution path (*Pesic & van der Aalst, 2006*; *Heinl et al., 1999*).

*Vasilecas, Kalibatiene & Lavbič (2016)* propose distinguishing BP modeling approaches according to *the levels of dynamicity*. The first and lowest level of dynamicity is described as using *decision points*, in which a human or an automated system decides what to do next according to predefined rules. Almost all of today's approaches and tools implement this level of dynamicity. The second, middle level of dynamicity is understood as the ability to *automatically configure* BP, like choosing an alternative template or paths for processing activities, or changing activities or their execution order in a process when context changes. Most of today's approaches fall within this level of dynamicity, a good example of which is the use of variable models (*Rosa et al., 2017*; *Alférez et al., 2014*; *Milani et al., 2016*). The third and highest level of dynamicity is achieved in the case of, for example, a *goal-driven* BP that can be changed at run-time according to the new conditions and the customer's needs, as in *Vasilecas et al. (2016)*. Therefore, these BP have no predefined sequence of activities. Summing up the defined dynamicity levels, they depend on the ability to create a BP instance according to the predefined BP model, or conversely to define specific activities and their sequences at run-time.

## Approaches to achieving dynamicity of BP

All of the processes mentioned previously (i.e., ad-hoc, agile, flexible, adaptive, variable, customizable, declarative, and dynamic) can be context-sensitive, rule-based, event-based, policy-based, case-handled, etc. According to *Saidani & Nurcan (2006)*, *Nunes, Werner & Santoro (2011)*, a context-aware, or *context-sensitive*, process is able to adapt BP instances at run-time to the changing context, which is defined by the minimum values of the variables that contain all relevant information that impacts the design and execution of a BP (*Rosemann & Recker, 2006*). In *Coutaz et al. (2005)*, a context is a set of entities, a set of roles that entities may satisfy, a set of relations between the entities, and a set of situations that denote specific configurations of entities, roles, or relations. Moreover, context can be internal and external. An internal context is represented by the state of resources in a system, such as the availability raw materials, and an external context is represented by the state of a system environment, such as the price of raw materials in the market, as defined by a set of variables. Existing approaches allow for the definition of one type of context individually: internal, as with the current state of system resources in

*Hu, Wu & Chen (2014)*, or external (*Saidani & Nurcan, 2006*; *Nunes, Werner & Santoro, 2011*; *Mejia Bernal et al., 2010*) only, without considering both.

In *rule-based* BP, rules can describe different aspects by ensuring some level of BP dynamicity. For example, the authors use rules to define the relationships between events, which caused during BP execution when changing a state of resources, and to filter the interesting ones in their proposed CEVICHE architecture (*Hermosillo, Seinturier & Duchien, 2010*). When an event that is important for BP adaptation is detected, the Complex Event Processing (CEP) engine notifies the responsible component to adapt the BP (instance or model) with the corresponding activity at run-time. *Milanovic, Gasevic & Rocha (2011)* use BR patterns to enrich BP in terms of possible cases and to increase BP flexibility. *Pesic & van der Aalst (2006)* use constraints to define relationships among tasks in their proposed ConDec language. At every moment during the execution of a process model, there is a judgement about whether or not the model satisfies the defined constraints. In *Mejia Bernal et al. (2010)*, the authors propose deconstructing BP into an ECA (Event-Condition-Action) rule set and adapting it to the external context data, describing user's priorities to provide a service. The authors describe expressing transitions between activities in the form of ECA rules, where an event is generated when an activity has finished its execution, a condition used to verify which workflow part is enabled, and where a rule action determines the next activity that has to be executed. The adaptation at run-time is then performed. All processes have a strictly defined sequence of activities, and this sequence of activities can be changed according to the defined set of rules (*Mejia Bernal et al., 2010*).

Here, based on the related works, we can summarize that BRs can ensure the dynamicity of a BP as follows: (1) each activity in a process is selected according to the defined conditions at BP run-time, and (2) the content of an activity is chosen based on the changing internal and/or external context. However, the rule-based approaches reviewed above do not cover both of these aspects.

In some papers (*Xiao et al., 2011*), process dynamicity is ensured through predefined *process fragments* and the defining of their relationships, which are specified in a constraint-based way. The *policies* contain a set of rules to select, whereby concrete fragment implementations will be used in the BP. Processes are dynamically generated based on constraints and adaptation policies according to their operating environments. Process fragments provide a modularized view on process models (*Xiao et al., 2011*). Consequently, the process can be changed at run-time by adding, substituting, or removing fragments, or modifying their relationships, which allows the process to be adapted dynamically. In some cases, this *policy-based* approach is similar to a rule-based approach, since a process schema is composed from fragments, the relationships between which are specified by the predefined rules. In addition, other researchers propose to ensure some level of BP dynamicity through an *event-based* approach (*Hermosillo, Seinturier & Duchien, 2010*; *Hermosillo, 2012*), which is discussed previously, as it uses rules to combine events into one complex one. Those complex events are necessary to notify the responsible component, which in turn searches for the corresponding aspect to adapt the BP.

In *van der Aalst, Weske & Grünbauer (2005)*, the authors proposed a *case handling* paradigm for supporting flexible and knowledge intensive BP, which are based on data. As the authors state, case handling focuses on what can be done to achieve a business goal. The central concepts in a case handling approach are the case and its data as opposed to the activities and routing rules in traditional workflow[1] management systems. Usually, case-handled systems present all data about a case at any time to the user. In such systems, many cases can be handled in parallel and these cases are logically independent. According to *Mutschler, Weber & Reichert (2008)*, in case-handling a system is more flexible compared to a workflow management system, since there is no error-prone "context tunneling" as in workflow based approaches. The case-handling paradigm is considered as "a more flexible approach" because users work with whole cases and can, for different reasons, modify the predefined process model (*van der Aalst, Weske & Grünbauer, 2005*).

*Sabatucci & Cossentino (2019)* views dynamic workflows as a promising approach to provide flexible BP execution in the dynamic business environment. Their main contribution is an automated procedure to extract implicit goals from a BPMN workflow description. This study is highly dependent on BPMN workflows, functional dimension of workflows, and availability of services, which implement workflows. Comparing, in this review, we concentrate on dynamic changes at the business level, but not at the implementation level, as presented in *Sabatucci & Cossentino (2019)*.

### The definition of a dynamic BP (DBP)

This research focuses on different approaches to achieving BP dynamicity. Therefore, we present a study allowing us to determine what level of dynamicity is achieved by various approaches. Moreover, for our survey we are going to use the definition of a dynamic BP (DBP) as follows.

**Definition 1:** A **dynamic business process** (DBP) is a process that is able to support structural and functional changes (i.e., has no predefined activities nor sequence of activities) at DBP instance run-time according to its context and rules, and that can be implemented with minimal delay.

A **context** refers to an internal context and an external context of DBP. As outlined in a number of papers, an external context is a set of variables and context rules defining a particular state of the environment. If the current state of the environment changes, then the external context also changes. An internal context is the current state of system resources. If the current state of system resources changes, then the internal context also changes.

According to the DBP definition, the DBP requirements (DBPR) adopted from *Vasilecas, Kalibatiene & Lavbič (2016)* are defined as the following:

DBPR-1. DBP should not have a predefined sequence of activities.

DBPR-2. DBP should react to the change of the context.

DBPR-3. Every subsequent activity should be selected according to predefined rules and a context. If there is no activity for further execution, it should be possible to do the following:

---

[1] As defined in *Zur Muehlen (2004)*, a workflow is a formal, or implementation-specific, representation of a BP.

3.1. to terminate the execution of a DBP instance; or

3.2. to define a new activity and related rules for a DBP instance execution.

DBPR-4. DBP changes can be initiated by any role involved, at any time, with possibly low latency compared to a DBP execution time.

DBPR-5. Before selecting the next activity, the historical data detailing instances of execution of the same DBP should be analyzed and the next activity selected should not cause the execution of an unacceptable sequence of activities, as defined by prior experience.

DBPR-6. DBP execution should align with a particular business goal.

We argue that DBP is a BP which meets all of the requirements presented. Implementing all six requirements provides more freedom in BP modeling. However, this freedom can be constrained by adding BRs, which depend on the requirements of the application domain. Moreover, the advantage of implementing the proposed requirements is that it allows DBP to be goal-oriented.

## RELATED WORKS

Using the research strategy presented in the "Survey methodology" section, we have identified a number of existing literature reviews on BP dynamicity. There are a number that align closely with our research (*Rosa et al., 2017*; *Pourshahid et al., 2012*; *Ayora et al., 2015*; *Cognini et al., 2014*; *Cognini et al., 2018*; *Kapuruge, Han & Colman, 2010*; *Goedertier, Vanthienen & Caron, 2015*).

*Rosa et al. (2017)* present a survey of papers on BP variability modeling by aiming to identify the commonalities and differences of approaches, criteria to select between different approaches, and research gaps that exist in the literature. As the authors identified, a standard process model is extended to capture process variants into a customizable process model. A model of each variant can be changed by adding or deleting process model fragments according to the context. The authors assessed their identified approaches using the 14 criteria described in their paper and produced a report that identified a number of gaps in these approaches. Firstly, they recognized a lack of effective methods and tools to support and assist users in the creation, use, and maintenance of the proposed approaches, leading to their limited adoption in practice. Another gap was that around half of the approaches reviewed have been validated through case studies. As the authors noticed, there is a lack of comparative empirical evaluations with end-users, which might provide evidence that one variability modeling approach is more usable than others in a particular setting. Though the paper presents very detailed analysis of the state-of-the-art, the authors do not present their survey method, which differs from others presented in literature reviews, such as (*Ivarsson & Gorschek, 2011*). Moreover, there is no access to the appendices of the research, where some important information about the survey is presented. Therefore, not all of the criteria that is outlined can be understood completely.

*Pourshahid et al. (2012)* present SLR on aspect-oriented approaches for BP adaptation. They have observed that current methods focus on the following: (1) composing and

swapping services based on Quality of Service, cost, rules, policies, and constraints, as well as in the event of failure; (2) extracting roles and cross-cutting concerns from composite services; (3) customizing process instances based on user profiles or Service Level Agreements; (4) adapting service composition and collaboration policies; and (5) using monitoring aspects to detect undesired situations. As a result of the review, authors proposed their own aspect-oriented process modeling and adaptation framework, which considers organizational goals, performance, and constraints as a whole when improving BP. The main limitation of the (*Pourshahid et al., 2012*) study is that it focuses on service-based BP adaptation, i.e., implementation level of BP.

*Ayora et al. (2015)* present an evaluation of variability support in process-aware information systems (PAIS). Based on the results of their literature review, the authors presented the VIVACE framework, which allows for the systematic assessment and comparison of existing approaches to process variability. As the authors state, VIVACE enables process engineers to select the variability approach that best meets their requirements, as well as helping them in implementing PAIS supporting process variability. Specifically, VIVACE comprises a core set of variability-specific language constructs and a core set of features that foster process variability along the process lifecycle. These constructs allow the assessment of the expressiveness of existing process variability approaches regarding the modeling of process variability. Despite the advantage of the (*Ayora et al., 2015*) study and the detailed analysis of BP, this study is limited to variable BP only.

*Cognini et al. (2014)* present a literature review on BP adaptation by aiming to find what raises the need for adaptation within the BP domain, which BP life cycle management phases require support for adaptation, which are the instruments used to express and support BP adaptation, whether there are any real experiences of BP adaptation, and what the challenges associated with the BP adaptation are. Whilst the authors have identified a large amount of interest in the analised topic, there are some gaps left in the research, namely the lack of application of research results to real BP adaptation scenarios. They identified just three real BP adaptation scenarios: the car logistic of the seaport of Bremen (Germany) (*Bucchiarone, Mezzina & Pistore, 2013*), the warehouse management (*Marconi et al., 2009*), and the clinical (*Dadam & Reichert, 2009*) scenarios. Other main issues associated with the BP adaptation include: the need for dynamic languages for BP modeling; current languages for BP modeling failing to support adaptation constructs; the adaptation of BP running instances; the verification of adapted BP; and evolving BP.

In *Kapuruge, Han & Colman (2010)*, authors have analyzed BP flexibility in the scope of service composition according to the following criteria: (1) process definition flexibility according to the possibility of BP modifications during the analysis and design phases; (2) process instance flexibility according to the possibility of process instance change at run-time; (3) services relationship flexibility according to a service composition in terms of the relationships among the different entities of the composition. During the survey, the

authors found that there is a lack of support for modeling service relationships, which describes the mutual obligations, constraints, etc., in defining the BP of a service composition. Compared to the current study (*Kapuruge, Han & Colman, 2010*) focuses on and is limited to the BP flexibility in the scope of service composition.

*Cognini et al. (2018)* present a literature review on BP flexibility with a focus on software systems related aspects. They define flexibility as the ability to properly manage the coordination between challenges in organizational aspects and technical environments, and their changes. The authors' literature review is based on the guidelines presented in *Kitchenham (2007)*. The results obtained have confirmed the increasing relevance of the BP flexibility topic, aided by the authors' analysis of papers on BP flexibility over a broader timescale, ranging from 2000 to 2015. As the authors have found, flexibility issues influence all BP life-cycle stages. Whilst there are a number of approaches proposed to support BP flexibility, there is still a lack of application of the proposed approaches in concrete and complex cases. Moreover, the authors identified several research directions in this field that require further investigation. They are as follows: modeling languages for flexibility implementation in BP; verification of the flexible BP model; adaptation of BP instances at run-time; and evolving BP.

In *Goedertier, Vanthienen & Caron (2015)*, authors focused on the literature review of declarative BP modeling principles and languages. They have chosen eight declarative process modeling approaches and languages with distinct declarative specifications, such as BPCN (*Lu, Sadiq & Governatori, 2009*), ConDec (*Pesic & van der Aalst, 2006*), etc. These approaches are compared according to the state space, which is a specific configuration of the facts about entities (e.g., BP activities) in a state space corresponding to a specific context of a BP, the transition types among states, and rules (e.g., transition rules). As the authors stated, the approaches that they analyzed differ mainly on the matter of interests, because approaches were developed to represent some reality, as with the ConDec language, which expresses BRs only about sequence and its items' timing constraints. Other differences lie in a differing understanding of a state space, constraint types expressing different ways of transitions, and knowledge representation and reasoning paradigms, which use additional different ontologies. Moreover, as the authors have concluded, flexibility costs money, and therefore the efficiency of executing declarative models may be complicated. This is because of the extended possibilities of choosing suitable execution paths, managing BRs, etc.

In drawing this together, it is evident that the surveys that were analyzed examine BP dynamicity from different perspectives. These include: variability (*Rosa et al., 2017*; *Ayora et al., 2015*); aspect-oriented and services based (*Pourshahid et al., 2012*); service compositions (*Kapuruge, Han & Colman, 2010*); flexibility (*Cognini et al., 2014*; *Cognini et al., 2018*); and declarative (*Goedertier, Vanthienen & Caron, 2015*). Therefore, the results obtained and conclusions drawn by the authors are limited to their particular respective areas. Consequently, there is a need for more general studies in the area of BP dynamicity. In this study, however, we are not going to compete with existing studies

but to supplement them in terms of both time and the scale of their scope. The main advantage of the current study is that it provides the analysis of a DBP meaning from a global perspective.

## SURVEY METHODOLOGY

In this section, a research method is presented that begins with a brief literature review of papers published on DBPs, proceeds to a detailed review of the refined set of papers, and ends with conclusions. The research method schema is presented in Fig. 1, and is adapted from (*Ivarsson & Gorschek, 2011*; *Kitchenham, 2007*; *Kitchenham et al., 2009*).

### The list of searching sources

As the study is focused on BP dynamicity, i.e., BP modeling approaches and their implementation into software systems, relevant papers should be searched in databases covering Computer Science (CS), Information Systems (IS), and Software Engineering (SE). Due to technical limitations only free access sources were chosen, of which there were a limited number (see Table 1). However, the initial study of sources shows that they contain a significant number of papers relevant to the research questions.

### A search query and keywords

According to the scope of the survey, keywords and a search query were defined. First, we defined the main keywords taking into account synonyms and terms related to each of the three concepts, as shown in Table 2.

Other search terms–such as agile, ad-hoc, or customizable–were not used as keywords for the search. Adding those terms increased the number of papers found, but additional papers did not expand the knowledge provided on the research topic. They were excluded from the search but are still found together with the keywords used. The scope of the survey shows that our search query results cover the topic of BP dynamicity. The proximity operator, i.e., *, was used to find more relevant results by allowing variations of the same terms, for example business process and business processes, dynamic and dynamicity, dynamical, etc.

Secondly, the queries were combined from the keywords using Boolean operators as follows.

**SQ1:** ("dynamic* business process*" OR "business process* dynamic*" OR "dynamic* in business process*" OR "dynamic* BP" OR "BP dynamic*" OR "dynamic* in BP")

**SQ2:** ("flexib* business process*" OR "business process* flexib*" OR "flexib* in business process" OR "flexib* BP" OR "BP flexib*" OR "flexib* in BP")

**SQ3:** ("variab* business process*" OR "variab* in business process*" OR "business process* variab*" OR "variab* BP" OR "variab* in BP" OR "BP variab*")

**SQ4:** ("adapt* in business process*" OR "business process* adapt*" OR "adapt* business process*" OR "adapt* in BP" OR "BP adapt*" OR "adapt* BP")

The results of the search according to the defined queries are presented in Table 3.

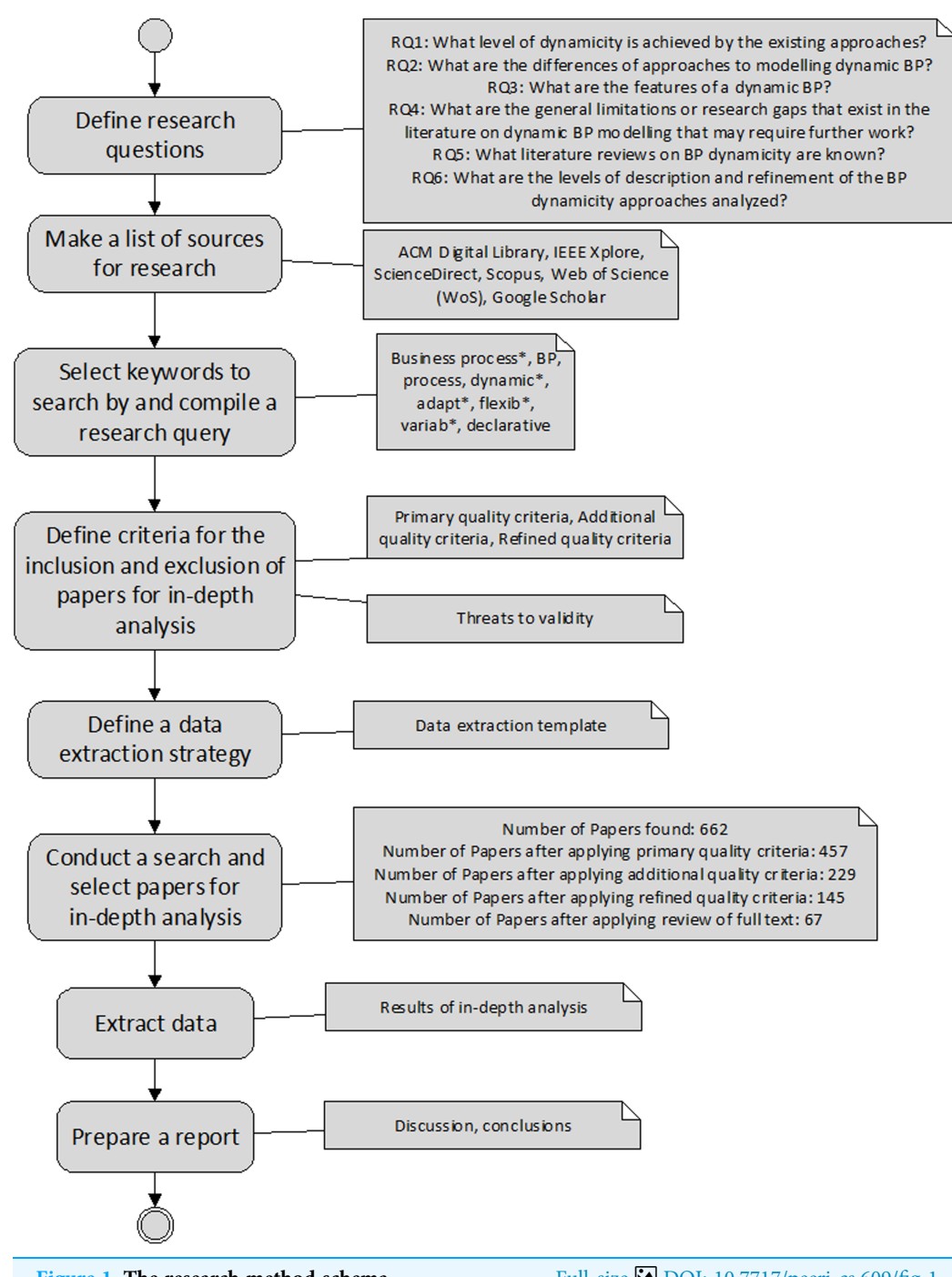

**Figure 1 The research method schema.**

The search in the databases was restricted to title, abstract, and keywords, except in WoS where it was done according to the topic (the other option was to search a title or a full text separately). Google Scholar is more a search engine than a digital database. Therefore, we used it to supplement the search results and to reduce the threats caused by the specifics of some databases.

**Table 1  List of sources.**

| Data source | Website |
|---|---|
| 1. ACM Digital Library | https://dl.acm.org |
| 2. IEEE Xplore | http://ieeexplore.ieee.org/Xplore/home.jsp |
| 3. ScienceDirect | http://www.sciencedirect.com/ |
| 4. Scopus | https://www.scopus.com/ |
| 5. Web of Science (WOS) | https://apps.webofknowledge.com/ |
| 6. Google Scholar | https://scholar.google.lt |

**Table 2  Keywords used for the search.**

| Main concept | Related terms | Keywords used in the search |
|---|---|---|
| Business process | BP | Business process*, BP, process |
| Dynamic | Dynamicity, dynamism, dynamics, dynamically, dynamical, adaptability, adaptation, adaptable, adapt, adaptive, flexible, flexibility, variable, variability declarative | Dynamic*<br>Adapt*<br>Flexib*<br>Variab*<br>Declarative |

**Note:**
The proximity operator "*" was used to find more relevant results by allowing variations of the same terms.

**Table 3  Results of the performed search (May, 2019).**

| Data source | SQ1 | SQ2 | SQ3 | SQ4 | Total: |
|---|---|---|---|---|---|
| 1. ACM Digital Library | 15 | 7 | 6 | 13 | 41 |
| 2. IEEE Xplore | 50 | 27 | 2 | 58 | 137 |
| 3. ScienceDirect | 25 | 20 | 9 | 13 | 67 |
| 4. Scopus | 120 | 118 | 41 | 94 | 373 |
| 5. Web of Science (WoS) | 10 | 15 | 5 | 14 | 44 |
| Total: | 220 | 187 | 63 | 205 | 662 |

Other important primary quality criteria were applied during the search. Publications that featured the following restrictions were excluded from the results obtained:

1. A publication that is non-peer-reviewed–including books, Master's/PhD theses, keynotes, tutorials and editorials, etc.
2. A publication that is not available in the English language.
3. Two or several papers that are the same. Duplicates of papers were excluded using Mendeley (https://www.mendeley.com), since some papers are included into several databases and repeated.

After applying a set of *primary quality criteria*, the number of papers decreased from 662 to 457.

However, evidently there were still too many papers to perform an analysis. Therefore, an additional refinement was applied to the results in order to decrease the number of papers, and to ensure their relevance to the analyzed topic.

As a result, the titles and abstracts of the papers were additionally reviewed to exclude those that were deemed irrelevant. The *additional quality criteria* used were defined as follows:

1. Exclude papers which contain relevant keywords, but within which DBP analysis is not the main topic of the paper, as in the fields of: education, medicine, energy efficiency, trade-off, fault identification, economic, artificial intelligence, programming, and hardware.
2. Exclude papers with a length of less than six pages, as such short papers present only a general idea and omit details of an overall approach.

After the refinement of the papers obtained in the earlier stages, we were left with 229 papers to be studied in further detail.

## Study selection

From the title and abstract of a paper, only incomplete information can be gathered regarding the context and the proposed approach presented in the study. Therefore, the remaining 229 papers were reviewed in more detail, manually applying the following steps and deciding to include or exclude the paper:

1. Read the abstract of the paper.
2. Read the introduction to the paper.
3. If a decision to include or exclude the paper is not made, read the discussion and conclusions.
4. If it is still unclear, search for the keywords in the text of the paper and evaluate their contextual suitability to the scope of our study.

Moreover, the refined quality criteria for selecting papers were as follows:

1. The paper should present a method or a technique on the subject of BP dynamicity.
2. The paper should not present a language without its application in a case study.
3. The paper should present an evaluation, which can be understood as: an application, a test or illustration, a toy example, a case study, an experiment, or any sort of empirical evaluation.

After the application of the detailed review process, 145 papers remained for review via the reading of the full text. After reading the full texts of these papers, 67 papers were selected for data extraction and detailed analysis.

## Threats to validity

The papers for analysis were chosen based on the strategy described previously, which included: (1) searching sources; (2) selection via keywords and queries; and (3) selection

via quality criteria. All of these variabilities were selected based on the related works analysis in the corresponding BP dynamicity area. However, there exists the possibility of missing important studies, because not all papers can be located using defined keywords that are related to the research questions. In order to reduce these threats to validity, the following measures were taken:

- A manual checking of the references in the extracted papers was patiently carried out in order to identify those papers that might be missed during the initial search using the keywords that we had defined. In addition, a precise definition of the paper selection quality criteria that complied with the research questions was enforced to avoid the incorrect exclusion of suitable papers.
- The snowballing search method, applied from *Jalali & Wohlin (2012)*, was used on the found literature reviews on BP dynamicity approaches. We went forward by reviewing the reference lists of these literature reviews by identifying papers cited in the reviews and assessing them according to our defined study selection criteria. We have discovered additional ten papers meeting our inclusion criteria via this snowballing procedure.
- Reading the abstract and the title of the papers introduces a threat, because the abstract and the title allows the exclusion of papers that are not relevant at the first glance. Even if it did not necessarily reflect what is actually presented in the papers, this threat allowed us to exclude papers that were definitely not relevant (*Ivarsson & Gorschek, 2011*).
- Reading the full-text of papers and evaluating them according to the defined paper selection criteria allowed us to limit a set of relevant papers and exclude papers that possessed defined keywords, but that were not relevant to the area of BP dynamicity that was analyzed.
- The sources searched (see Table 1) were limited to the most relevant sources, and included only peer-reviewed papers. Note that Google Scholar was used as a supplementary search engine.

Finally, to minimize the threat associated with inaccurate extraction of data, the following threats to validity were used:

- Including only papers from journals and proceedings classified as CS, IS, and SE categories limits the possibility of generalizing the results to other forums, in which SE technologies are published.
- Primary, additional, and refined paper extraction criteria were used to select papers.
- All of the selected papers were assessed using the questions presented in Table 2.

## Data extraction strategy

In this step, data were extracted from the selected relevant papers regarding BP dynamicity according to the features defined in Table 4, and presented in the table rows. Each feature could be clearly understood in the context of BP dynamicity after answering one or several questions, presented within a feature. These features and questions are based on

**Table 4 Data extraction template according to the requirements of DBP.**

| RQ | Process dynamicity influencing features that are mentioned in analyzed papers | Possible answers | Scale | Ranking (RQ1) | Rigor (RQ6) |
|---|---|---|---|---|---|
| | Meta information: study ID and reference of a paper. | | | | |
| RQ1 RQ2 RQ3 RQ4 | 1. Process model predefined (essential) (see DBPR-1):<br>1.1. Does a process have a predefined sequence of activities, i.e., a predefined process model?<br>1.2. Is a process based on meta-model, general model, generic model, or abstract model?<br>1.3. Is it possible to define additional activities and to change or delete existing activities or skip to other activities at process instance run-time? | Yes/No<br>Yes/No<br>Yes/Part/No | 0/1<br>0/1<br>1/ 0.5/ 0 | X<br>X<br>X | |
| RQ1 RQ2 RQ3 RQ4 | 2. Context aware (Essential) (see DBPR-2):<br>2.1. Does an external context affect the execution of the process?<br>2.2. Does an internal context affect the execution of the process?<br>2.3. Is a process instance affected by the change of a context at run-time? | Yes/No<br>Yes/No<br>Yes/No | 1/0<br>1/0<br>1/0 | X<br>X<br>X | |
| RQ1 RQ2 RQ3 RQ4 | 3. Rule-based[1] (Essential) (see DBPR-3):<br>3.1. Is every subsequent activity in a process selected according to the predefined rules at process instance run-time?<br>3.2. Is it possible to define new rules, or change or delete existing rules, at process instance run-time?<br>3.3. Does process instance react to the rule changes made at process instance run-time?<br>3.4. Are rules used to solve other tasks? | Yes/Part/No<br>Yes/No<br>Yes/No<br>Yes/No | 1/ 0.5/ 0<br>1/0<br>1/0<br>1/0 | X<br>X | |
| RQ1 RQ2 RQ3 RQ4 | 4. Changes (Essential): Are changes implemented at process instance run-time with low latency? (see DBPR-4) | Yes/No | 1/0 | X | |
| RQ1 RQ2 RQ3 RQ4 | 5. Accumulation of experience (see DBPR-5):<br>5.1. Is the experience of each process instance execution stored?<br>5.2. Is it possible to classify executed instances of a process as a "good practice" and a "bad practice"?<br>5.3. Are time, cost, etc. values calculated and stored for each process instance?<br>5.4. Is it possible to restrict the execution of process instances that are named "bad instances"?<br>5.5. Is historical data[3] used for solving other related tasks? | Yes/No<br>Yes/No<br>Yes/No<br>Yes/No | 1/0<br>1/0<br>1/0<br>1/0 | X<br>X<br>X | |
| RQ1 RQ2 RQ3 RQ4 | 6. Goal: Is a process goal-oriented? (see DBPR-6) | Yes/No | 1/0 | X | |
| RQ1 RQ2 RQ3 RQ4 | 7. Ontology: Is domain ontology or process ontology used in an approach? | Yes/No | 1/0 | X | |
| | Completeness of description of process dynamicity approach and its implementation | | | | |
| RQ2 RQ3 RQ4 RQ6 | 8. Approach description:<br>8.1. Does the paper present a clear description of the proposed approach on BP dynamicity?<br>8.2. Does the paper present a description of the proposed approach on dynamic BP in a formal language? | Yes/Part/No<br>Yes/Part/No | 1/ 0.5/ 0<br>1/ 0.5/ 0 | | X<br>X |
| RQ1 RQ2 RQ3 RQ4 RQ6 | 9. Dynamicity implementation approach (if 1.1. is yes):<br>9.1. Is the process dynamicity realized through variants?[2]<br>9.2. Is the process dynamicity realized through cases?<br>9.3. Is the process dynamicity realized through a declarative approach?<br>9.4. Is the process dynamicity realized through goal-orientation?<br>9.5. Is the process dynamicity realized through other approaches? | Yes/No<br>Yes/No<br>Yes/No<br>Yes/No<br>Yes/No | 1/0<br>1/0<br>1/0<br>1/0<br>1/0 | X<br>X<br>X<br>X<br>X | X<br>X<br>X<br>X<br>X |
| RQ2 RQ3 RQ4 RQ6 | 10. Implementation of the proposed approach:<br>10.1. Is any case of implementation of a proposed model presented in the paper?<br>10.2. What approach, language or technique is used for the implementation of the proposed approach?<br>10.3. Does the paper present a case study or a prototype or a descriptive example?[4]<br>10.4. Does the paper present a clear description of an implementation with text, screen shots, and code lines?<br>10.5. Does the paper evaluate[5] the results obtained? | Yes/No<br>Answer<br>Yes/Part/No<br>Yes/Part/No<br>Yes/Part/No | 1/0<br>1/ 0.5/ 0<br>1/ 0.5/ 0<br>1/ 0.5/ 0 | X | X<br>X<br>X<br>X |

(Continued)

| RQ | Process dynamicity influencing features that are mentioned in analyzed papers | Possible answers | Scale | Ranking (RQ1) | Rigor (RQ6) |
|---|---|---|---|---|---|
| RQ4 RQ6 | 11. Discussion and conclusions:<br>11.1. Does the paper present an appropriate discussion on dynamicity?<br>11.2. Does the paper present grounded conclusions about dynamicity? | Yes/Part/No<br>Yes/Part/No | 1/<br>0.5/<br>0<br>1/<br>0.5/<br>0 | | X<br>X |

**Notes:**

[1] Here a rule role is analyzed in a process. In analyzed approaches, rules are used for different tasks (3.4), as with transformation rules that are used to generate the process variants from patterns in *Abderrahmane, Mili & Boubaker (2014)*, etc. In some approaches, like *Bucchiarone et al. (2017)*, the rule concept is not used at all. Authors use "transition relations," which in general can be termed rules.

[2] A variant-based process here means that process change can be made in predefined variation points. A case-based process definition is the same as in *van der Aalst, Weske & Grünbauer (2005)*.

[3] Historical data is used in some a way, for example a log file used to build a model (*Mattos et al., 2014*), etc.

[4] If a paper presents a case study or a prototype, then the answer is *yes*. If a paper presents a descriptive example, then the answer is *part*. If a paper does not present any implementation, then the answer is *no*.

[5] This means that some type of validation, i.e., Experts or other, is presented here.

dynamic BP requirements presented in *Vasilecas, Kalibatiene &Lavbič (2016)*. If a feature is not explicitly mentioned in the paper, it is mapped according to this survey analysts' understanding. In some cases, a paper can describe several studies. Therefore, the primary study is selected according to the scope of the survey.

Table 4 consists of four columns, where the first column presents research questions related to the feature. The second column presents features and questions. The third column presents possible answers to the questions. As can be seen, there are two types of questions. First are those that concern the approach, and elicit a yes or no answer. Second are those that concern either an approach or a description of the approach, eliciting a yes, part, or no answer. The fourth column represents the scaling of answers, where one means a strong description (i.e., a feature is described to the degree where a reader can understand and compare it to the same aspect in another paper), 0.5 represents a medium description (i.e., a feature in which the study performed is mentioned or presented in brief, but not described to the degree to which a reader can clearly understand and compare it to the same feature in other paper), and zero where there is a weak description or indeed none at all.

At the end of the examination of the approach, presented in the paper on the basis of scaling, a ranking, denoting the level of dynamicity of an approach and based on features 1–7, 9, and 10 from Table 4, and a rigor, denoting the level of description of an approach and based on features 8–11 from Table 4, are summed up. Thus, all papers are ranked according to their BP dynamicity. The larger the ranking value, the larger the dynamicity level that is achieved by the proposed approach. Also, the larger the rigor value, the larger the level of description, i.e., the better quality of approach, is achieved in the paper. Other examination features, such as problem definition, etc., can be added to Table 4. However, they are neglected intentionally in order to optimize assessment procedure. We consider that the global aim of those studies is the same.

The results of the data extraction are presented in the "Results" section.

**Table 5 Assessment questions for evaluating existing literature reviews.**

| No. | Questions | Answers | Scale | Relevance | Rigor |
|---|---|---|---|---|---|
| Q1 | Whether the aims of the analyzed survey and this survey match? | Yes/Part/No | 1/0.5/0 | X | |
| Q2 | Whether the sources used in the analyzed survey and in this survey match? | Yes/Part/No | 1/0.5/0 | X | |
| Q3 | Whether the meaning of research questions in the analyzed survey and in this survey similar? | Yes/Part/No | 1/0.5/0 | X | |
| Q4 | Whether the search keywords in the analyzed survey and this survey cover the similar field of research? | Yes/Part/No | 1/0.5/0 | X | |
| Q5 | Is the analyzed survey based on literature review guidelines agreed in software engineering? | Yes/No | 1/0 | X | X |
| Q6 | Is the analyzed survey based on authors' defined approach? (if previous question is No) | Yes/No | 0.5/0 | X | X |
| Q7 | Does the analyzed survey identifies and mitigates its threats to validity? | Yes/Part/No | 1/0.5/0 | X | X |
| Q8 | Does the analyzed survey include a quality assessment of the performed search? | Yes/Part/No | 1/0.5/0 | X | X |
| Q9 | Does the analyzed survey include a quality assessment of primary studies? | Yes/Part/No | 1/0.5/0 | X | |
| Q10 | Are the results presented in the analyzed survey described properly? | Yes/Part/No | 1/0.5/0 | | X |
| Q11 | Does the analyzed survey have described examples of flexibility or variability? | Yes/Part/No | 1/0.5/0 | X | X |
| Q12 | Does the analyzed survey have answered to the defined research questions? | Yes/Part/No | 1/0.5/0 | X | X |
| Q13 | Does the analyzed survey have presented evident conclusions? | Yes/Part/No | 1/0.5/0 | | X |
| Maximum value | | | 12 | 11 | 8 |
| Minimum value | | | 0 | 0 | 0 |

## Comparison of literature reviews on BP dynamicity

Among the papers presenting BP dynamicity approaches, papers with literature reviews were also found. For the comparison of those papers we have developed assessment criteria which is based on questions presented elsewhere (*Cognini et al., 2018*; *Ivarsson & Gorschek, 2011*; *Kitchenham et al., 2009*), and shown in Table 5. In the table, the first column presents the number of a question, the second presents the question itself, the third presents possible answers to the questions, the fourth presents the scale or value of an answer, the fifth notes whether the question belongs to the relevance category, and the sixth notes whether the question belongs to the rigor category. Thus, in this paper, each literature review is assessed according to the questions presented and, based the answers provided, relevance, denoting the level of similarity of an analyzed survey to the present survey, and rigor, denoting the quality and presentation of an analyzed survey, are calculated. The scaling of answers is then the same as in Table 4.

## RESULTS

The main results obtained during the survey of the selected papers consist of three main parts as follows: (1) surveys on the topic of BP dynamicity; (2) papers proposing

new BP dynamicity approaches; and (3) analysis of tools for DBP modeling and simulation.

## Comparison of literature reviews on BP dynamicity (RQ5)

A summary of the results, obtained during the deep analysis of literature reviews according to the questions from Table 5, is presented in Tables 6 and 7.

As can be seen from Table 7, though, the publication years of the analyzed surveys on BP dynamicity differs. The main period of the papers analyzed is in the range from 2000–2015, and the refined and overlapped period of analyzed papers is in the range from 2004–2013. The surveys analyzed and the present survey generated a set of about 500 research papers, which concern research questions on BP dynamicity. All surveys have analyzed BP dynamicity from different perspectives as follows: BP variability (*Rosa et al., 2017*; *Ayora et al., 2015*); aspect oriented (*Pourshahid et al., 2012*); service compositions (*Kapuruge, Han & Colman, 2010*); BP flexibility (*Cognini et al., 2014*; *Cognini et al., 2018*); and declarative BP (*Goedertier, Vanthienen & Caron, 2015*). The surveys most similar to this survey are (*Cognini et al., 2014*; *Cognini et al., 2018*). As the survey presented in *Cognini et al. (2018)* is newer and more sophisticated than (*Cognini et al., 2014*), it was selected to compare with this survey according to the sources used. This comparison shows that 58% of the sources analyzed in this survey differ from the sources used in *Cognini et al. (2018)*. Such partial overlapping can be explained by the similarity of the topic analyzed. However, as the analysis of our references shows, this survey covers the most up to date papers on BP dynamicity, including the year 2018, which could not have been be analyzed in another survey under review because they only covered papers in the period of 2000–2015. Moreover, the research questions defined in both surveys differ, and distinctions can also be seen in keywords. One additional keyword used in our search query was "dynamic" and its variations. The primary difference among this survey and that presented by *Cognini et al. (2018)* lies in different paper selection criteria, i.e., exclusion and inclusion criteria. In this survey only the most comprehensive and non-duplicated papers by the same authors were chosen for detailed analysis, but not all papers from the same authors.

Another interesting and important aspect observed during the comparison of the surveys analyzed is the quality assessment of the search that was undertaken to identify relevant papers. Only two surveys from those analyzed have presented such an assessment. In *Rosa et al. (2017)*, two authors assessed each approach independently, and third author combined the results, after which confirmation was finally sought from the authors of each primary paper. In *Cognini et al. (2018)*, a quite different assessment of the selected papers was employed. The data collected were validated by three experts with more than 10 years of experience in BP management, along with other related expertise. The feedback from experts was reconsidered by the authors in a meeting to consolidate the final version of the questions. Additionally, a validation based on the snowballing search method (*Jalali & Wohlin, 2012*) was used on the selected research papers before the next steps in the analysis. Other authors do not present any assessment procedures.

**Table 6 Summary of reviews on DBP.**

| Reference | Scope | Data sources | Period | Methodology | Data extraction | Quality assessment | Threats to Validity |
|---|---|---|---|---|---|---|---|
| (Rosa et al., 2017) | BP variability by restriction and by extension | Concentration not on sources, but on particular approaches | 2000–2014 | Not presented | 66 relevant papers: 23 approaches, out of which 11 main approaches subsume the other 12 approaches | Not presented | Not presented |
| (Pourshahid et al., 2012) | Aspect-oriented approaches for BP adaptation | One source, not specified | 2000–2012 | Presented, usual to literature reviews | 56 papers | Quality criteria for selecting papers | (1) Searched on article titles and only one meta-search engine used (2) Looking at commonly cited articles by the authors of the papers found, and by comparing the authors' conclusions with some more generic surveys and studies |
| (Ayora et al., 2015) | Evaluation of variability support in process-aware IS | SpringerLink, IEEE Xplore Digital Library, ACM Digital Library, Science Direct – Elsevier, Wiley Inter Science, World Scientific | 2004–2013 | Presented, usual to literature reviews | 63 papers | Quality criteria for selecting studies | Selection bias, inaccuracy in data extraction and analysis, and reliability |
| (Cognini et al., 2014) | BP adaptation | IEEExploer, ACM Digital Library, Citeseerx Library, ScienceDirect, Springer Link, Web Of Science | 2000–2013 | Presented, usual to literature reviews | 84 papers | Three inclusion criteria and four exclusion criteria | Not presented |
| (Cognini et al., 2018) | BP flexibility with a focus on software systems | IEEExplorer, ACM Digital Library, Citeseerx Library, Scopus Science Direct, SpringerLink, WOS | 2000–2015 | The method proposed in (Kitchenham, 2007) | 164 papers | Two inclusion criteria, two exclusion criteria, and seven questions developed by (Dyba & Dingsoyr, 2008) | A validation based on the snowballing search method (Jalali & Wohlin, 2012) |
| (Goedertier, Vanthienen & Caron, 2015) | Declarative process modeling languages and principles | Concentration not on sources, but on particular approaches | Languages of period 1998-2009 | A set of criteria is chosen | 8 languages | Not discussed | Not discussed |
| (Kapuruge, Han & Colman, 2010) | Process flexibility in the context of service compositions | Not presented | Not presented | Not presented | 7 papers | Not described | Not presented |
| This study | BP dynamicity and flexibility | ACM Digital Library, IEEE Xplore, ScienceDirect, Scopus, WOS | 2005–2017 | The method proposed in (Ivarsson & Gorschek, 2011) | 60 papers | Three stage exclusion procedure, described in Section 4 | Presented in Section 3.5 |

**Table 7  An initial data on literature reviews comparison.**

| Reference | Q1 | Q2 | Q3 | Q4 | Q5 | Q6 | Q7 | Q8 | Q9 | Q10 | Q11 | Q12 | Q13 | Relevance | Rigor |
|---|---|---|---|---|---|---|---|---|---|---|---|---|---|---|---|
| (Rosa et al., 2017) | 0.5 | 0.5 | 0.5 | 0.5 | 0 | 1 | 0 | 1 | 0.5 | 0.5 | 1 | 0.5 | 1 | 6 | 5 |
| (Pourshahid et al., 2012) | 0.5 | 0 | 0 | 0.5 | 1 | 0 | 1 | 0 | 1 | 1 | 0 | 0 | 1 | 4 | 4 |
| (Ayora et al., 2015) | 0.5 | 0 | 0 | 0.5 | 1 | 0 | 1 | 0 | 1 | 1 | 1 | 1 | 0.5 | 6 | 5.5 |
| (Cognini et al., 2014) | 1 | 1 | 0.5 | 1 | 1 | 0 | 0 | 0 | 1 | 1 | 0 | 1 | 1 | 6.5 | 4 |
| (Cognini et al., 2018) | 1 | 1 | 0.5 | 1 | 1 | 0 | 0.5 | 1 | 1 | 1 | 0 | 1 | 1 | 8 | 5.5 |
| (Kapuruge, Han & Colman, 2010) | 0.5 | 0 | 0 | 0 | 0 | 1 | 0 | 0 | 0 | 0.5 | 1 | 0 | 1 | 2.5 | 3.5 |
| (Goedertier, Vanthienen & Caron, 2015) | 0.5 | 0 | 0 | 0.5 | 0 | 1 | 0 | 0 | 0 | 1 | 1 | 1 | 1 | 4 | 5 |
| SUM | 4 | 2.5 | 1.5 | 3.5 | 4 | 2 | 2.5 | 2 | 4.5 | 5 | 3 | 3.5 | 5.5 | 33 | 27.5 |
| AVG | 0.643 | 0.357 | 0.214 | 0.571 | 0.571 | 0.429 | 0.357 | 0.286 | 0.643 | 0.857 | 0.571 | 0.643 | 0.929 | 5.286 | 4.643 |
| MIN | 0.500 | 0.000 | 0.000 | 0.000 | 0.000 | 0.000 | 0.000 | 0.000 | 0.000 | 0.500 | 0.000 | 0.000 | 0.500 | 2.500 | 3.500 |
| MAX | 1.000 | 1.000 | 0.500 | 1.000 | 1.000 | 1.000 | 1.000 | 1.000 | 1.000 | 1.000 | 1.000 | 1.000 | 1.000 | 8.000 | 5.500 |

Table 7 presents data obtained from the answering of the questions from Table 5. As can be seen from the results, the maximum relevance have achieved by Cognini et al. (2018) (8 points). This can be explained by the fact that it is the most similar to the present survey. The differences between both surveys were discussed previously in this section. The maximum rigor of 5.5 points is achieved by Ayora et al. (2015) and (Cognini et al., 2018), denoting that the presentation of surveys is at a high level in their papers.

In Fig. 2, we have presented an overview of rigor-ranking scores for the literature reviews on BP dynamicity. As can be seen, the reviews analyzed have achieved middle or high levels of sophistication in presenting the results of analysis on the topic of BP dynamicity. However, some gaps in reviews still remain. They are as follows: presenting a research methodology (Q5 and Q6) (Ayora et al., 2015); threats to validity (Q7) (Rosa et al., 2017; Cognini et al., 2014; Kapuruge, Han & Colman, 2010; Goedertier, Vanthienen & Caron, 2015); and quality assessment of the performed search (Q8 and Q9) (Ayora et al., 2015; Pourshahid et al., 2012; Cognini et al., 2014) (see Fig. 3).

## Comparison of papers on BP dynamicity

Here we present the results of comparison between papers on BP dynamicity. After the primary review of the full texts of papers, a set of 67 papers remained for in-depth analysis. It should be noted that similar papers from the same authors were merged by choosing the most complete and up to date paper. Figure 4 presents the number of papers on BP dynamicity selected for in-depth analysis varying by years. As can be seen from this figure, the number of papers on BP dynamicity rises slightly. The results of papers' assessments according to the predefined features of BP dynamicity (Table 4) is presented in Table 8. They are discussed below by proceeding through each column and answering the previously defined research questions. The categorization of the analyzed papers

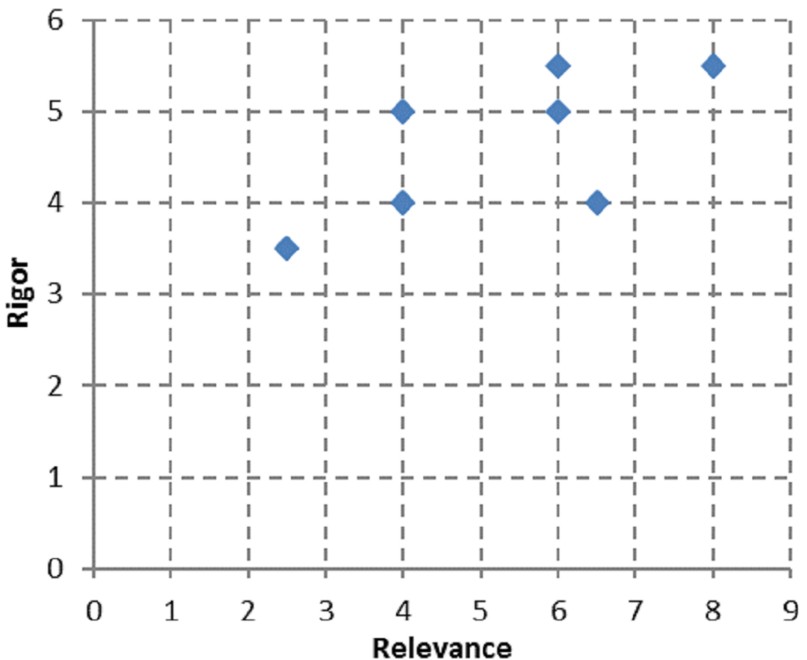

**Figure 2 An overview of rigor-ranking scores for the papers on literature reviews on BP dynamicity.**

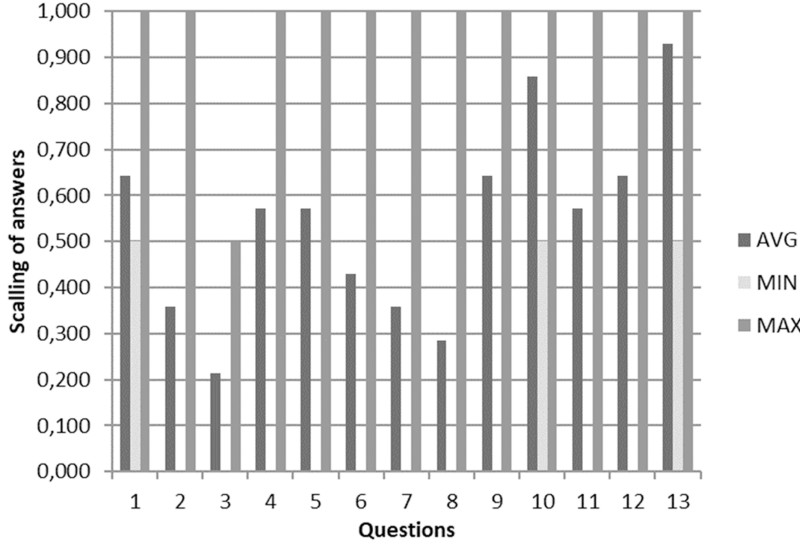

**Figure 3 An overview of answers on the assessment of the papers on literature reviews on BP dynamicity.** An overview of answers on the assessment of the papers on literature reviews on BP dynamicity.

according to the process dynamicity influencing features (Table 4) are presented in "Appendix 1".

In Table 8, the first three columns (1.1–1.3) represent the dependency of all process instances from a predefined BP model, i.e., 1.1–1.3 correspond to the features from Table 4 that influence process dynamicity. According to whether the process model is predefined

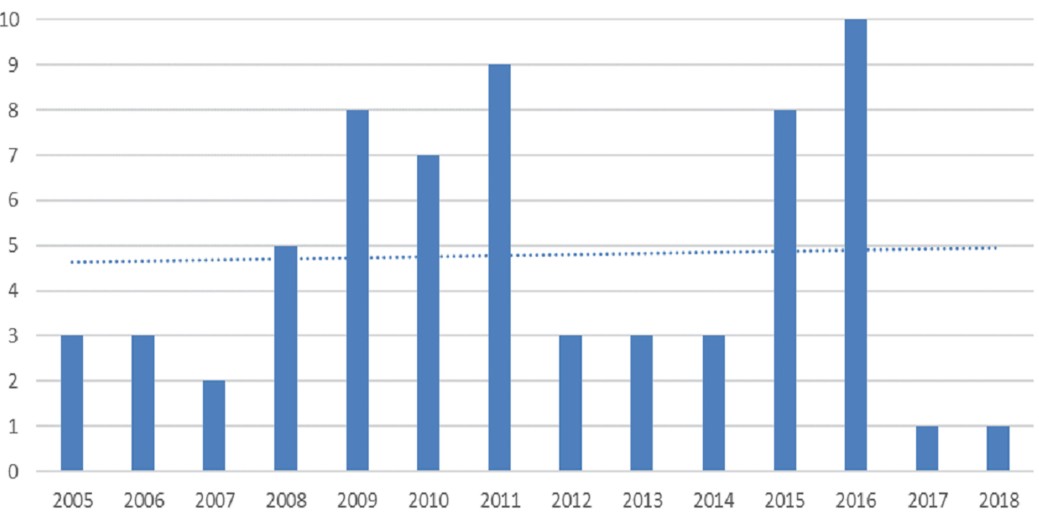

**Figure 4 The number of papers on BP dynamicity selected for in-depth analysis by year.**

or not, in 88% of the papers analyzed (59 papers), approaches consider BP dynamicity through the change of the existing process model by using: process variants, which are implemented by the restriction or extension of activities in a process instance (*Rosa et al., 2017*; *Ayora et al., 2016*; *Hallerbach, Bauer & Reichert, 2010*); generic process model (*Abderrahmane, Mili & Boubaker, 2014*) or process meta-model (*Mattos et al., 2014*), which is used to generate a process; or process cases (*Sprovieri et al., 2016*), when possible activities are known in advance (i.e., at design-time), but their sequencing is defined at run-time depending on the context. Other parts of the approaches analyzed (12%: eight papers) consider that there is no predefined process model, i.e., the process is made up of process fragments (*Haarmann et al., 2015*) or composed of rules, like ECA rules in *Mejia Bernal et al. (2010)*, which are selected at process instance run-time according to the context. Moreover, in only 45% (30 papers) of approaches is it possible to modify activities at process instance run-time.

The next three columns (2.1–2.3) in Table 8, which correspond Table 4, allow us to investigate the extent to which a process is dependent on the context, which can change at run-time. Columns 2.1 and 2.2 indicate the type of a context, i.e., external (technological evolution, new working methods, changes in laws, changes in the target goal, exceptions, economic requirements, etc. (*Cognini et al., 2018*)) or internal (such as an organization's resources). In about half of these papers, an external (51%) or internal (51%) context is considered, and in other part of papers context is not considered at all. In 27% both external and internal contexts are considered. In 58% (39 papers) of approaches, a process instance can react to the change in an external and/or internal context at run-time.

The third set of columns (3.1–3.4) shows the use of rules in approaches. As can be seen from column 3.1, 82% (55 papers) of approaches use rules (*van Eijndhoven, Iacob & Ponisio, 2008*; *Mejia Bernal et al., 2010*; *Santo Carvalho et al., 2013*) or policies (*Cao et al., 2009*) to select the next activity in a process instance. However, only 9% (six papers) feature authors discussing changes in rules (3.2) and reacting to those changes (3.3) during process

**Table 8 Initial data on BP dynamicity.**

| Reference | 1.1 | 1.2 | 1.3 | 2.1 | 2.2 | 2.3 | 3.1 | 3.2 | 3.3 | 3.4 | 4 | 5.1 | 5.2 | 5.3 | 5.4 | 5.5 | 6 | 7 | 8.1 | 8.2 | 9.1 | 9.2 | 9.3 | 9.4 | 9.5 | 10.1 | 10.3 | 10.4 | 10.5 | 11.1 | 11.2 | Ranking | Rigor |
|---|---|---|---|---|---|---|---|---|---|---|---|---|---|---|---|---|---|---|---|---|---|---|---|---|---|---|---|---|---|---|---|---|---|
| (Abderrahmane, Mili & Boubaker, 2014) | 0 | 0 | 1 | 1 | 1 | 0 | 1 | 0 | 0 | 1 | 1 | 1 | 0 | 0 | 0 | 1 | 0 | 0 | 1 | 0 | 0.5 | 0 | 0 | 0 | 0 | 1 | 1 | 1 | 1 | 1 | 0 | 9.5 | 6.5 |
| (van der Aalst, Weske & Grünbauer, 2005) | 1 | 0 | 0.5 | 1 | 1 | 1 | 1 | 0 | 0 | 0 | 1 | 0 | 0 | 0 | 0 | 0.5 | 0 | 0 | 1 | 1 | 0 | 0.5 | 0 | 0 | 0 | 1 | 1 | 1 | 0.5 | 0 | 1 | 8.5 | 7.0 |
| (Asuncion, Iacob & van Sinderen, 2010) | 0 | 1 | 0 | 0 | 0 | 0 | 1 | 0 | 0 | 1 | 0 | 0 | 0 | 0 | 0 | 0 | 1 | 1 | 1 | 0 | 0 | 0 | 0 | 1 | 0 | 1 | 1 | 1 | 0 | 0 | 1 | 6.0 | 6.0 |
| (Awadid & Gnannouchi, 2015) | 0 | 1 | 1 | 0 | 0 | 0 | 0 | 0 | 0 | 0 | 1 | 0 | 0 | 0 | 0 | 0 | 0 | 0 | 0.5 | 0 | 0 | 0 | 0 | 0 | 0.5 | 1 | 0 | 0.5 | 0 | 0 | 1 | 4.5 | 3.5 |
| (Ayora et al., 2016) | 0 | 1 | 1 | 1 | 1 | 1 | 1 | 0 | 0 | 0 | 1 | 1 | 0 | 0 | 0 | 0 | 0 | 0 | 1 | 0 | 0.5 | 0 | 0 | 0 | 0 | 1 | 1 | 1 | 0.5 | 1 | 0 | 7.5 | 6.0 |
| (Ayora et al., 2012) | 0 | 1 | 0 | 0 | 1 | 1 | 1 | 0 | 0 | 0 | 1 | 0 | 0 | 0 | 0 | 0 | 1 | 0 | 1 | 0 | 0.5 | 0 | 0 | 0.5 | 0 | 1 | 1 | 1 | 0 | 0 | 1 | 9.0 | 6.0 |
| (Boffoli et al., 2012) | 0 | 0 | 0 | 1 | 1 | 0 | 1 | 0 | 0 | 0 | 0 | 0 | 0 | 0 | 0 | 0 | 0 | 0 | 0 | 0 | 0.5 | 0 | 0 | 0 | 0 | 1 | 1 | 1 | 0 | 0 | 1 | 4.5 | 5.5 |
| (Boffoli, Cimitile & Maggi, 2009) | 0 | 0 | 0 | 1 | 0 | 0 | 1 | 0 | 0 | 1 | 0 | 0 | 0 | 0 | 0 | 0 | 0 | 0 | 1 | 0 | 0.5 | 0 | 0 | 0 | 0 | 1 | 1 | 1 | 0.5 | 0 | 1 | 3.5 | 7.0 |
| (Bögl, Natschläger & Geist, 2016) | 0 | 0 | 0 | 1 | 1 | 1 | 1 | 0 | 0 | 1 | 1 | 1 | 0 | 1 | 1 | 1 | 0 | 0 | 1 | 0 | 0.5 | 0 | 0 | 0 | 0 | 1 | 1 | 0.5 | 0 | 0 | 1 | 10.5 | 5.0 |
| (Borrego & Barba, 2014) | 1 | 0 | 0 | 0 | 0 | 0 | 1 | 0 | 0 | 0 | 0 | 1 | 0 | 1 | 0 | 1 | 1 | 1 | 1 | 0 | 0 | 0 | 1 | 0 | 0 | 1 | 1 | 1 | 0 | 1 | 1 | 9.0 | 8.0 |
| (Bucchiarone, Mezzina & Pistore, 2013) | 1 | 0 | 1 | 1 | 0 | 1 | 1 | 0 | 0 | 1 | 1 | 0 | 0 | 0 | 0 | 0 | 1 | 1 | 1 | 1 | 0.5 | 0 | 0 | 0 | 0 | 1 | 1 | 1 | 0 | 0 | 1 | 8.5 | 6.5 |
| (Bucchiarone et al., 2011) | 0 | 1 | 1 | 1 | 1 | 0 | 1 | 0 | 0 | 1 | 1 | 1 | 0 | 0 | 0 | 1 | 1 | 1 | 1 | 0 | 0 | 0 | 0 | 0.5 | 0 | 1 | 1 | 1 | 1 | 0 | 1 | 10.5 | 6.5 |
| (Cao et al., 2009) | 0 | 1 | 1 | 0 | 1 | 1 | 1 | 0 | 0 | 0 | 1 | 1 | 0 | 0 | 0 | 1 | 0 | 0 | 1 | 0 | 0 | 0 | 0 | 0 | 0.5 | 1 | 1 | 1 | 0 | 0 | 1 | 9.5 | 5.5 |
| (Châtel, Malenfant & Truck, 2010) | 0 | 1 | 0 | 0 | 1 | 0 | 1 | 0 | 0 | 0 | 0 | 0 | 0 | 0 | 0 | 0 | 0 | 1 | 1 | 0 | 0.5 | 0 | 0 | 0 | 0 | 1 | 1 | 1 | 0 | 0 | 1 | 6.5 | 5.5 |
| (Cherif, Djemaa & Amous, 2015) | 0 | 1 | 0 | 1 | 0 | 1 | 0 | 0 | 0 | 0 | 1 | 1 | 0 | 0 | 0 | 0 | 0 | 0 | 1 | 0 | 0.5 | 0 | 0 | 0 | 0 | 1 | 1 | 1 | 0 | 0 | 1 | 5.5 | 5.5 |
| (Czepa et al., 2016) | 0 | 1 | 1 | 0 | 1 | 0 | 1 | 1 | 1 | 0 | 1 | 0 | 0 | 0 | 0 | 0 | 0 | 1 | 1 | 0 | 0 | 0.5 | 0 | 0 | 0 | 1 | 1 | 1 | 1 | 1 | 1 | 11.5 | 7.5 |
| (Dadam & Reichert, 2009) | 0 | 0 | 1 | 0 | 1 | 1 | 1 | 0 | 0 | 1 | 1 | 1 | 0 | 0 | 0 | 0 | 0 | 0 | 1 | 0 | 0.5 | 0 | 0 | 0 | 0 | 1 | 1 | 1 | 0 | 0 | 0 | 7.5 | 4.5 |
| (Deb, Chaki & Ghose, 2015) | 0 | 1 | 0 | 1 | 0 | 0 | 0 | 0 | 0 | 0 | 0 | 0 | 0 | 0 | 0 | 0 | 1 | 0 | 1 | 1 | 0 | 0 | 0 | 1 | 0 | 1 | 1 | 1 | 1 | 0 | 1 | 6.0 | 8.0 |
| (Demeyer et al., 2010) | 0 | 0 | 1 | 0 | 0 | 1 | 1 | 0 | 0 | 0 | 1 | 1 | 0 | 0 | 0 | 0 | 0 | 1 | 0 | 1 | 0 | 0 | 1 | 0 | 0 | 1 | 1 | 1 | 0 | 0 | 1 | 7.0 | 7.0 |
| (Gong & Janssen, 2012) | 0 | 1 | 0 | 0 | 0 | 0 | 0 | 0 | 0 | 0 | 1 | 0 | 0 | 0 | 0 | 0 | 1 | 1 | 0 | 0 | 0 | 0 | 0.5 | 1 | 0 | 1 | 0.5 | 0.5 | 0 | 1 | 1 | 7.5 | 6.5 |
| (Haarmann et al., 2015) | 0 | 0 | 0 | 0 | 0 | 0 | 1 | 0 | 0 | 0 | 1 | 1 | 0 | 1 | 0 | 1 | 0 | 0 | 1 | 0 | 0 | 0.5 | 0 | 0 | 0 | 0 | 1 | 1 | 0 | 0 | 1 | 7.5 | 5.5 |
| (Hallerbach, Bauer & Reichert, 2010) | 0 | 0 | 0 | 1 | 0 | 1 | 1 | 0 | 0 | 0 | 1 | 1 | 0 | 0 | 0 | 0 | 0 | 0 | 1 | 1 | 0.5 | 0 | 0 | 0 | 0 | 1 | 1 | 1 | 0 | 0 | 1 | 5.5 | 7.5 |
| (Hermosillo, Seinturier & Duchien, 2010) | 0 | 1 | 1 | 1 | 0 | 1 | 1 | 0 | 0 | 0 | 1 | 1 | 0.5 | 0 | 0 | 0 | 0 | 0 | 1 | 0 | 0.5 | 0 | 0 | 0 | 0 | 1 | 0.5 | 0.5 | 1 | 1 | 1 | 9.0 | 7.5 |
| (Jiang et al., 2016) | 0 | 0 | 0 | 1 | 1 | 1 | 1 | 0 | 0 | 0 | 1 | 1 | 0 | 0 | 0 | 0 | 0 | 0 | 1 | 1 | 0.5 | 0 | 0 | 0 | 0 | 1 | 1 | 1 | 0 | 0 | 1 | 7.5 | 7.5 |
| (Kannengiesser et al., 2014) | 0 | 0 | 0 | 0 | 0 | 0 | 0 | 0 | 0 | 0 | 0 | 0 | 0 | 0 | 0 | 0 | 0 | 1 | 1 | 0 | 0.5 | 0 | 0 | 0 | 0.5 | 1 | 1 | 0 | 0 | 0 | 1 | 4.5 | 6.5 |
| (Kapuruge, Han & Colman, 2011) | 0 | 0 | 1 | 0 | 0 | 0 | 1 | 0 | 0 | 0 | 1 | 1 | 0 | 0 | 0 | 0 | 1 | 1 | 1 | 1 | 0 | 0 | 0 | 0 | 0.5 | 1 | 1 | 1 | 0 | 0 | 1 | 5.5 | 6.5 |
| (Khriss et al., 2008) | 1 | 1 | 1 | 0 | 1 | 1 | 0 | 0 | 0 | 0 | 1 | 1 | 0 | 0 | 0 | 0 | 0 | 0 | 0 | 0 | 0 | 0 | 0 | 0 | 0.5 | 1 | 1 | 1 | 0 | 0 | 1 | 7.5 | 5.5 |
| (Kim, Kim & Kim, 2007) | 0 | 1 | 1 | 0 | 0 | 1 | 1 | 0 | 0 | 0 | 1 | 1 | 0 | 0 | 0 | 0 | 0 | 0 | 0 | 0 | 0.5 | 0 | 0 | 0 | 0 | 1 | 1 | 1 | 0 | 0 | 1 | 7.5 | 5.5 |
| (Li & Du, 2016) | 0 | 0 | 0 | 0 | 0 | 0 | 0 | 0 | 0 | 0 | 0 | 0 | 0 | 0 | 0 | 0 | 0 | 0 | 0 | 0 | 0.5 | 0 | 0 | 0 | 0 | 0.5 | 0.5 | 0.5 | 0 | 0 | 1 | 2.5 | 5.5 |
| (Liu et al., 2012) | 1 | 1 | 1 | 0 | 1 | 1 | 1 | 0 | 0 | 0 | 1 | 1 | 0 | 0 | 0 | 0 | 0 | 0 | 1 | 0 | 0 | 0 | 0 | 0 | 0.5 | 1 | 1 | 1 | 0 | 0 | 1 | 8.5 | 6.5 |
| (Marconi et al., 2009) | 0 | 0 | 1 | 1 | 1 | 1 | 0 | 0 | 0 | 0 | 1 | 0 | 0 | 0 | 0 | 0 | 0 | 1 | 1 | 0.5 | 0.5 | 0 | 0 | 0 | 0 | 1 | 1 | 1 | 0 | 0 | 1 | 7.5 | 6.0 |

(Continued)

| Reference | 1.1 | 1.2 | 1.3 | 2.1 | 2.2 | 2.3 | 3.1 | 3.2 | 3.3 | 3.4 | 4 | 5.1 | 5.2 | 5.3 | 5.4 | 5.5 | 6 | 7 | 8.1 | 8.2 | 9.1 | 9.2 | 9.3 | 9.4 | 9.5 | 10.1 | 10.3 | 10.4 | 10.5 | 11.1 | 11.2 | Ranking | Rigor |
|---|---|---|---|---|---|---|---|---|---|---|---|---|---|---|---|---|---|---|---|---|---|---|---|---|---|---|---|---|---|---|---|---|---|
| (Martinho, Domingos & Varajão, 2015) | 0 | 0 | 0 | 0 | 0 | 0 | 1 | 0 | 0 | 1 | 1 | 1 | 0 | 0 | 0 | 0 | 0 | 0 | 1 | 1 | 0.5 | 0 | 0 | 0 | 0 | 1 | 1 | 1 | 0 | 0 | 1 | 5.5 | 6.5 |
| (Mattos et al., 2014) | 0 | 0 | 1 | 1 | 0 | 1 | 0.5 | 0 | 0 | 0 | 1 | 1 | 0 | 0 | 0 | 1 | 0 | 0 | 1 | 1 | 0 | 0 | 0 | 1 | 0 | 1 | 1 | 1 | 1 | 1 | 1 | 8.5 | 9.0 |
| (Mejia Bernal et al., 2010) | 0 | 1 | 1 | 1 | 1 | 1 | 1 | 1 | 1 | 0 | 1 | 0 | 0 | 0 | 0 | 0 | 1 | 1 | 0 | 0 | 0 | 0 | 0.5 | 0 | 0 | 1 | 1 | 1 | 0.5 | 0 | 1 | 11.5 | 6.0 |
| (Mounira & Mahmoud, 2010) | 0 | 1 | 1 | 1 | 1 | 1 | 1 | 0 | 0 | 0 | 1 | 1 | 0 | 0 | 0 | 1 | 0 | 0 | 1 | 0 | 0 | 0 | 0 | 0 | 0.5 | 1 | 1 | 1 | 0 | 0 | 1 | 10.5 | 5.5 |
| (Natschläger et al., 2016) | 0 | 0 | 1 | 0 | 1 | 0 | 1 | 0 | 0 | 0 | 0 | 0 | 0 | 0 | 0 | 0 | 0 | 0 | 0 | 0 | 0.5 | 0 | 0 | 0 | 0 | 1 | 1 | 1 | 0 | 0 | 1 | 4.5 | 5.5 |
| (Oberhauser, 2016) | 0 | 1 | 0 | 0 | 1 | 1 | 1 | 0 | 0 | 0 | 1 | 0 | 0 | 0 | 0 | 0 | 0 | 0 | 0 | 0 | 0.5 | 0 | 0 | 0 | 0 | 1 | 1 | 0.5 | 0 | 0 | 1 | 6.5 | 5.0 |
| (Prasad et al., 2015) | 0 | 0 | 0 | 0 | 1 | 0 | 1 | 0 | 0 | 0 | 1 | 1 | 0 | 0 | 0 | 0 | 0 | 1 | 1 | 0 | 0 | 0 | 0 | 0.5 | 0 | 1 | 0.5 | 0.5 | 0 | 0 | 1 | 5.5 | 4.5 |
| (Ramakrishnan, 2009) | 0 | 0 | 0 | 0 | 1 | 0 | 0 | 0 | 1 | 0 | 1 | 0 | 0 | 0 | 0 | 0 | 0 | 1 | 1 | 0 | 0.5 | 0 | 0 | 0 | 0 | 1 | 0.5 | 0.5 | 0 | 0 | 1 | 6.5 | 4.5 |
| (Rong, Liu & Liang, 2008) | 1 | 1 | 0 | 1 | 1 | 1 | 1 | 0 | 0 | 0 | 1 | 1 | 0 | 0 | 0 | 0 | 0 | 0 | 1 | 0 | 0.5 | 0 | 0 | 0 | 0.5 | 1 | 0.5 | 0.5 | 0 | 0 | 1 | 8.5 | 4.5 |
| (Saidani & Nurcan, 2006) | 0 | 0 | 0 | 0 | 1 | 0 | 1 | 0 | 0 | 0 | 0 | 0 | 0 | 0 | 0 | 0 | 1 | 0 | 1 | 0 | 0 | 0 | 0 | 0 | 0 | 0 | 0 | 0 | 0 | 0 | 1 | 3.0 | 2.0 |
| (Santo Carvalho, Santoro & Revoredo, 2015) | 0 | 1 | 0 | 1 | 0 | 1 | 1 | 1 | 1 | 0 | 1 | 1 | 0 | 0 | 0 | 1 | 1 | 1 | 1 | 0 | 0 | 0 | 1 | 1 | 0 | 1 | 1 | 1 | 1 | 0 | 1 | 14.0 | 8.0 |
| (Santo Carvalho et al., 2013) | 0 | 1 | 0 | 1 | 1 | 0 | 1 | 0 | 0 | 0 | 0 | 0 | 0 | 0 | 0 | 0 | 1 | 1 | 0 | 1 | 0 | 0 | 0 | 0.5 | 0 | 1 | 1 | 1 | 0 | 0 | 1 | 8.5 | 5.5 |
| (Santos et al., 2013) | 0 | 0 | 1 | 1 | 0 | 0 | 1 | 0 | 0 | 1 | 1 | 0 | 0 | 0 | 0 | 0 | 0 | 0 | 0 | 0 | 0 | 0 | 1 | 0 | 1 | 1 | 1 | 1 | 0 | 0 | 1 | 7.0 | 7.0 |
| (Santos et al., 2011) | 0 | 1 | 0 | 1 | 0 | 1 | 0 | 0 | 0 | 0 | 1 | 1 | 0 | 0 | 0 | 0 | 0 | 0 | 1 | 0 | 0.5 | 0 | 0 | 0 | 1 | 1 | 0.5 | 0.5 | 0 | 0 | 1 | 6.5 | 4.5 |
| (Sarno et al., 2015) | 0 | 0 | 0 | 0 | 0 | 0 | 0 | 0 | 0 | 0 | 0 | 0 | 0 | 0 | 0 | 0 | 0 | 0 | 1 | 0 | 0.5 | 0 | 0 | 0 | 1 | 1 | 1 | 1 | 0 | 0 | 1 | 1.5 | 5.5 |
| (Sprovieri et al., 2016) | 0 | 0 | 0.5 | 1 | 1 | 1 | 1 | 0 | 0 | 0 | 1 | 1 | 0 | 0 | 0 | 0 | 1 | 0 | 1 | 0 | 0 | 0.5 | 0 | 0 | 0 | 1 | 0.5 | 0.5 | 0.5 | 0 | 1 | 8.0 | 5.0 |
| (Sun, Huang & Meng, 2011) | 0 | 1 | 0 | 0 | 1 | 1 | 1 | 0 | 0 | 0 | 1 | 1 | 0 | 0 | 0 | 0 | 0 | 0 | 1 | 1 | 0 | 0 | 0.5 | 0 | 1 | 1 | 1 | 1 | 0.5 | 0 | 1 | 6.5 | 7.0 |
| (Ukor & Carpenter, 2009) | 0 | 0 | 0 | 0 | 1 | 1 | 1 | 1 | 0 | 0 | 1 | 0 | 0 | 0 | 0 | 0 | 0 | 0 | 1 | 1 | 0.5 | 0 | 0 | 0 | 0 | 0 | 0 | 0 | 1 | 0 | 1 | 5.5 | 4.5 |
| (van Eijndhoven, Iacob & Ponisio, 2008) | 0 | 0 | 0 | 0 | 1 | 1 | 1 | 0 | 0 | 0 | 1 | 0 | 0 | 0 | 0 | 1 | 1 | 0 | 1 | 0 | 0.5 | 0 | 0 | 0 | 1 | 1 | 0.5 | 0.5 | 0 | 0 | 1 | 5.5 | 4.5 |
| (Wang & Capretz, 2007) | 1 | 0 | 0 | 0 | 0 | 0 | 1 | 0 | 0 | 0 | 1 | 1 | 0 | 0 | 0 | 0 | 1 | 1 | 1 | 1 | 0.5 | 0 | 0 | 0 | 0 | 1 | 1 | 1 | 0 | 0 | 1 | 5.5 | 6.5 |
| (Weidlich et al., 2011) | 0 | 0 | 0 | 0 | 0 | 0 | 1 | 0 | 0 | 1 | 0 | 0 | 0 | 0 | 0 | 0 | 0 | 0 | 1 | 0 | 0.5 | 0 | 0 | 0 | 0 | 1 | 1 | 1 | 0 | 0 | 1 | 3.5 | 6.5 |
| (Xia & Wei, 2008) | 0 | 1 | 1 | 1 | 1 | 1 | 0 | 0 | 0 | 0 | 1 | 1 | 0 | 0 | 0 | 0 | 1 | 1 | 1 | 1 | 0.5 | 0 | 0 | 0 | 0 | 1 | 1 | 1 | 1 | 1 | 1 | 10.5 | 7.5 |
| (Xiao et al., 2011) | 1 | 0 | 1 | 0 | 1 | 0 | 1 | 1 | 0 | 0 | 1 | 0 | 0 | 0 | 0 | 0 | 0 | 0 | 1 | 1 | 0 | 0 | 1 | 0 | 0 | 1 | 1 | 1 | 0 | 0 | 1 | 8.0 | 7.0 |
| (Yoo et al., 2008) | 0 | 0 | 0 | 1 | 1 | 1 | 1 | 1 | 1 | 0 | 1 | 0 | 0 | 0 | 0 | 0 | 0 | 0 | 1 | 1 | 0 | 0 | 0 | 0 | 0.5 | 1 | 1 | 1 | 0 | 0 | 1 | 8.5 | 6.5 |
| (Yousfi, Saidi & Dey, 2016) | 0 | 0 | 0 | 1 | 0 | 0 | 1 | 0 | 0 | 0 | 1 | 1 | 0 | 0 | 0 | 0 | 1 | 1 | 0 | 0 | 0.5 | 0 | 0 | 0 | 0 | 1 | 1 | 1 | 1 | 0 | 1 | 6.5 | 8.5 |
| (Yuliang et al., 2009) | 0 | 0 | 0 | 0 | 0 | 0 | 1 | 0 | 0 | 0 | 1 | 0 | 0 | 0 | 0 | 0 | 0 | 0 | 0 | 0.5 | 0 | 0 | 0 | 0.5 | 0.5 | 0.5 | 0.5 | 0.5 | 0 | 0 | 1 | 3.5 | 4.5 |
| (Gottschalk et al., 2008) | 0 | 0 | 1 | 0 | 0 | 1 | 1 | 0 | 0 | 0 | 1 | 0 | 0 | 0 | 0 | 0 | 0 | 0 | 0 | 0 | 0.5 | 0 | 0 | 0 | 0.5 | 1 | 1 | 1 | 0 | 0 | 1 | 4.5 | 5.5 |
| (Maggi et al., 2012) | 0 | 1 | 1 | 0 | 0 | 0 | 1 | 0 | 0 | 0 | 1 | 0 | 0 | 0 | 0 | 0 | 0 | 0 | 1 | 1 | 0 | 0 | 1 | 0 | 0 | 1 | 1 | 1 | 0 | 0 | 1 | 6.0 | 7.0 |
| (Pesic & van der Aalst, 2006) | 0 | 0 | 0 | 0 | 0 | 0 | 1 | 0 | 0 | 0 | 1 | 0 | 0 | 0 | 0 | 0 | 1 | 0 | 0 | 0 | 0 | 0 | 1 | 0 | 0.5 | 0.5 | 0 | 0.5 | 0 | 0 | 1 | 4.5 | 4.0 |
| (Wang & Wang, 2006) | 0 | 0 | 1 | 1 | 1 | 1 | 1 | 0 | 0 | 0 | 1 | 1 | 0 | 0 | 0 | 0 | 1 | 1 | 1 | 0.5 | 0 | 0 | 1 | 0.5 | 0 | 0.5 | 0.5 | 0.5 | 0 | 0 | 1 | 9.0 | 5.5 |
| (Baresi & Guinea, 2005) | 0 | 0 | 1 | 1 | 1 | 1 | 1 | 0 | 0 | 0 | 1 | 0 | 0 | 0 | 0 | 0 | 0 | 0 | 0 | 0 | 0 | 0 | 0 | 0 | 0.5 | 1 | 0.5 | 0.5 | 0 | 0 | 1 | 7.5 | 4.5 |
| (van der Aalst, Pesic & Schonenberg, 2009) | 0 | 1 | 0 | 0 | 0 | 0 | 1 | 0 | 0 | 0 | 1 | 0 | 0 | 0 | 0 | 0 | 0 | 0 | 1 | 0 | 0.5 | 0 | 1 | 0 | 0 | 1 | 1 | 1 | 0 | 0 | 1 | 5.0 | 6.0 |
| (Savickas & Vasilecas, 2018) | 1 | 1 | 1 | 1 | 1 | 1 | 1 | 0 | 0 | 0 | 1 | 1 | 0 | 0 | 0 | 1 | 1 | 1 | 1 | 0.5 | 0.5 | 0 | 0 | 1 | 1 | 1 | 1 | 1 | 0 | 0 | 1 | 13.0 | 6.5 |
| (Milanovic, Gasevic & Rocha, 2011) | 0 | 0 | 1 | 1 | 0 | 1 | 0 | 1 | 0 | 0 | 1 | 0 | 0 | 0 | 0 | 0 | 0 | 0 | 0 | 0 | 0.5 | 0 | 0 | 0 | 0 | 1 | 0.5 | 1 | 0 | 0 | 1 | 7.5 | 5.0 |
| (Bucchiarone et al., 2017) | 0 | 0 | 1 | 1 | 1 | 1 | 0.5 | 0 | 0 | 0 | 1 | 1 | 0 | 0 | 0 | 0 | 0 | 0 | 1 | 0.5 | 0 | 0 | 0 | 1 | 1 | 1 | 1 | 1 | 1 | 0 | 1 | 7.0 | 8.0 |
| (Vasilecas, Kalibatiene & Lavbič, 2016) | 1 | 1 | 1 | 1 | 1 | 1 | 1 | 1 | 0 | 1 | 1 | 1 | 0 | 0 | 1 | 0 | 0 | 1 | 0 | 0 | 0 | 0 | 0 | 0 | 0 | 1 | 1 | 1 | 1 | 0 | 1 | 17.0 | 7.0 |

instance run-time. In 30% (20 papers), rules are used for other purposes, such as transformation rules that are used to generate process variants from patterns in *Abderrahmane, Mili & Boubaker (2014)*.

The fourth column, in essence, repeats or summarizes whether the proposed approach allows changes (change of activities, change of context, and change of rules) in a process instance at run-time. We found that 82% (55 papers) of the papers analyzed consider changes at process instance run-time.

The fifth set of columns (5.1–5.5) presents the accumulation of experience of BP dynamicity. It was found that only 18% (12 papers) use historical data for some purpose, such as data mining, to generate possible process variables in *Abderrahmane, Mili & Boubaker (2014)*, or to build a model in *Mattos et al. (2014)*. Only 8% (five papers) calculate values, including time, cost, etc., for each executed process instance and use them to improve process results.

The sixth column considers whether a process is goal-oriented or not. As can be seen from the results obtained, in only 28% (19 papers) of cases is there mention of the goal-orientation of a process. We might conclude that other approaches assume that their processes are goal-oriented, but do not express as much explicitly.

The seventh column shows the usage of ontology in their approaches. Our research indicated that ontology is used for process model creation or knowledge about domain extraction in only 15% (10 papers) of approaches. The eighth set of columns (8.1 and 8.2) is used to present the completeness of the description of the proposed approach. As we found, all approaches are described in some way, and 37% (25 papers) of papers present a formal or semi-formal description of the proposed approach by using formulas.

The ninth set of columns (9.1–9.5) presents an approach to implementing BP dynamicity. In 31 papers, authors consider a variant-based approach, in four papers–a case-handling approach, in 11 papers–a declarative approach, in 12 papers–a goal-based approach, and in 11 papers–an "other" approach. These "other" approaches include a policy-based approach and a generic process model for workflow management (*Cao et al., 2009*), cognitive models and a function–behavior–structure (FBS) model (*Kannengiesser et al., 2014*), or service relationships (*Kapuruge, Han & Colman, 2011*). The tenth set of columns (10.1–10.5) is intended to describe the implementation perspective of approaches. As evidenced, 75% (50 papers) of papers had a case study and/or prototype description of their approaches. In 73% (49 papers), a clear description of an implementation with text, screen shots, and code lines was presented. Only 18% (12 papers) presented validation of the results obtained, whether from experts or other such sources.

The answers for question 10.2 are summarized below in Table 9, to avoid enlarging Table 8.

The analysis of question 10.2 demonstrates that the most popular architecture used for the implementation of DBP is SOA (49%, 33 papers of the 67 analyzed). Among popular languages for BP modeling are BPMN (22.39%, used in 15 papers) and BPEL (15%, used in ten papers). In five papers (7.46%), the authors do not provide any information about implementation, with the exception of some descriptive examples. Other implementation aspects are also interesting, as with the usage of rule engines, e.g., Jboss

**Table 9 Technology, language or other approaches used in the implementation section of the papers analyzed.**

| Reference | Technology, language, other approach | Results |
|---|---|---|
| (*Asuncion, Iacob & van Sinderen, 2010*; *Awadid & Gnannouchi, 2015*; *Bucchiarone et al., 2011*; *Châtel, Malenfant & Truck, 2010*; *Deb, Chaki & Ghose, 2015*; *Gong & Janssen, 2012*; *Hermosillo, Seinturier & Duchien, 2010*; *Kapuruge, Han & Colman, 2011*; *Khriss et al., 2008*; *Marconi et al., 2009*; *Oberhauser, 2016*; *Prasad et al., 2015*; *Ramakrishnan, 2009*; *Rong, Liu & Liang, 2008*; *Santos et al., 2011*; *Sun, Huang & Meng, 2011*; *Ukor & Carpenter, 2009*; *Wang & Capretz, 2007*; *Xia & Wei, 2008*; *Xiao et al., 2011*; *Yoo et al., 2008*; *Yuliang et al., 2009*; *Baresi & Guinea, 2005*; *Bucchiarone et al., 2017*) | SOA | 49% |
| (*van der Aalst, Weske & Grünbauer, 2005*; *Ayora et al., 2016*; *Bögl, Natschläger & Geist, 2016*; *Jiang et al., 2016*; *Marconi et al., 2009*; *Martinho, Domingos & Varajão, 2015*; *Mattos et al., 2014*; *Santos et al., 2013*; *Santos et al., 2011*; *van Eijndhoven, Iacob & Ponisio, 2008*; *Yousfi, Saidi & Dey, 2016*; *Milanovic, Gasevic & Rocha, 2011*) | BPMN | 15% |
| (*Cherif, Djemaa & Amous, 2015*; *Hallerbach, Bauer & Reichert, 2010*; *Hermosillo, Seinturier & Duchien, 2010*; *Kim, Kim & Kim, 2007*; *Marconi et al., 2009*; *Sun, Huang & Meng, 2011*; *Ukor & Carpenter, 2009*; *Yoo et al., 2008*; *Yuliang et al., 2009*; *Baresi & Guinea, 2005*) | BPEL | 10% |
| (*Abderrahmane, Mili & Boubaker, 2014*; *Asuncion, Iacob & van Sinderen, 2010*; *Sprovieri et al., 2016*) | MOF, MDA, CMMN | |
| (*Boffoli et al., 2012*; *Boffoli, Cimitile & Maggi, 2009*; *Mejia Bernal et al., 2010*) | Decision tables | 4.48% |
| (*Li & Du, 2016*; *Liu et al., 2012*; *Mejia Bernal et al., 2010*; *Saidani & Nurcan, 2006*; *Santo Carvalho, Santoro & Revoredo, 2015*) | Not provide any | 7.46% |

Drools 5.1 in *Abderrahmane, Mili & Boubaker (2014)*, Jess Java rule Engine (*Mounira & Mahmoud, 2010*; *Wang & Wang, 2006*), Goh with IBM WebSphere Business Modeler (*Ramakrishnan, 2009*). For other aspects of implementation see the "Implementation aspects of DBP" section.

The last set of columns (11.1 and 11.2) shows that 96% (64 papers) had conclusions, and 18% (12 papers) discussed their results.

According to Table 8, possible values of ranking are in the interval [0; 24] and possible values of rigor are in the interval [0; 13]. However, as can be seen from Table 8, Figs. 5, and 6, all of the papers analyzed fall within the ranking interval of [1.5; 17] and the rigor interval of [2; 9]. Moreover, the rigor of papers has been divided into intervals as follows:

- [2; 4)–low rigor, i.e., the paper has a poor description of an approach, an implementation, and a discussion with conclusions (two papers of 67, colored in blue).
- [4; 7)–middle rigor, i.e., the paper has a fair description of these elements (46 papers of 67, colored in red).
- [7; 9]–high rigor, i.e., the paper has strong a description of these elements (19 papers of 67, colored in green).

According to ranking, papers have been divided into intervals, also, as follows:

- [1; 7)–low ranking, i.e., the paper has a low level of dynamicity presented by their approach. In such papers the aspects analyzed (see Table 4) have weak descriptions or are not mentioned at all (30 papers of 67).

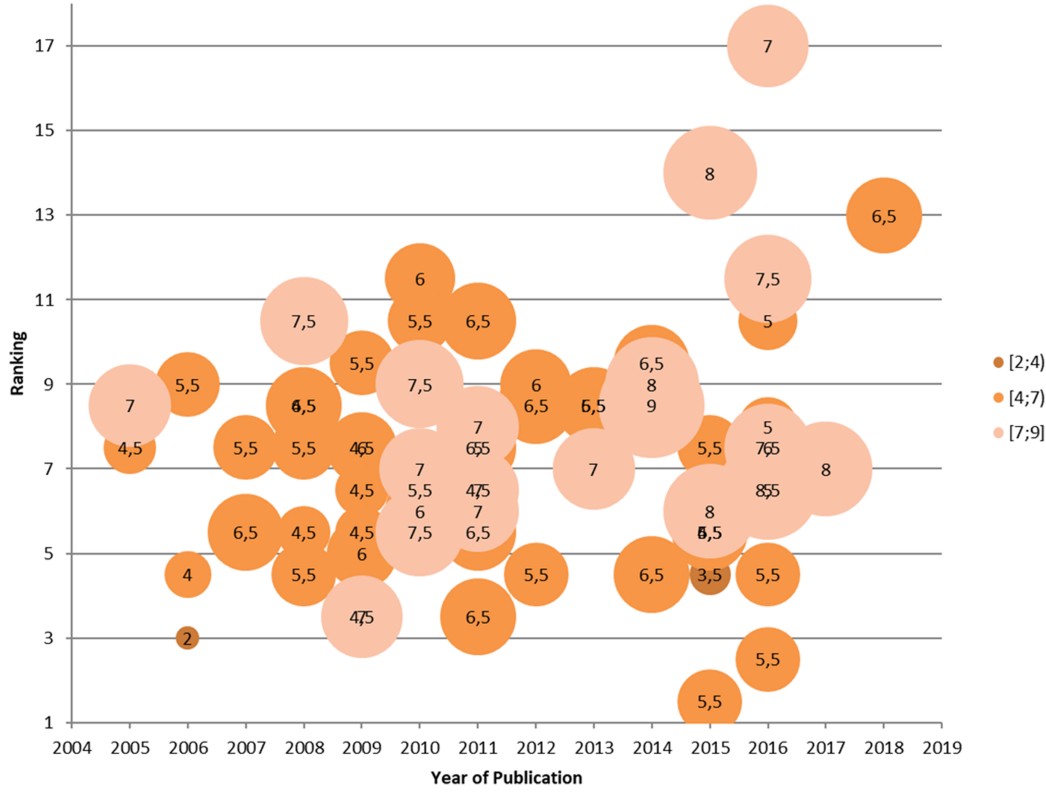

**Figure 5 Ranking of papers according to rigor.** Ranking of papers according to rigor.

- [7; 14)–middle ranking, i.e., the paper has a fair level of dynamicity presented by their approach. In such papers the aspects analyzed (see Table 4) have a fair description or some of the analyzed aspects, such as process model, context or rules, and implementation, have strong descriptions (35 papers of 67).

- [14; 22]–high ranking, i.e., the paper has a fair or strong description of all of the aspects analyzed (two papers of 67).

Summing up, the majority of papers have middle or high rigor and low or middle ranking (Figs. 5 and 6).

In Fig. 5, we present the ranking of papers, calculated based on the results presented in Table 8, according to the year of each papers' publication. The size of the bubbles depends on the rigor of papers per year. As can be seen from the figure, there are two periods–2008–2011 (30 publications) and 2014–2016 (22 publications)–when more papers on the topic of BP dynamicity were published.

The number of papers according to their rigor and ranking during the two identified periods (2008–2011 and 2014–2016) are presented in Table 10.

As can be seen from Table 10, in 2008–2011 half of the papers have high rigor (15 papers of 30). The remainder of the papers have low (12 papers of 30) or middle (three papers of 30) rigor in this period. In 2014–2016, almost half of the papers have high rigor (14 papers of 30). The remainder of the papers have low (two papers of 30) or middle

**Table 10 Number of papers according to their rigor and ranking during periods.**

| Rigor/ranking period | Low rigor [2; 4) | Middle rigor [4; 7) | High rigor [7; 9] | Low ranking [1; 7) | Middle ranking [7; 14) | High ranking [14; 22] | Number of papers |
|---|---|---|---|---|---|---|---|
| 2008–2011 | 3 | 12 | 15 | 15 | 15 | 0 | 30 |
| 2014–2016 | 2 | 9 | 14 | 11 | 9 | 2 | 22 |
| Number of papers | 5 | 11 | 29 | 26 | 24 | 2 | 55 |

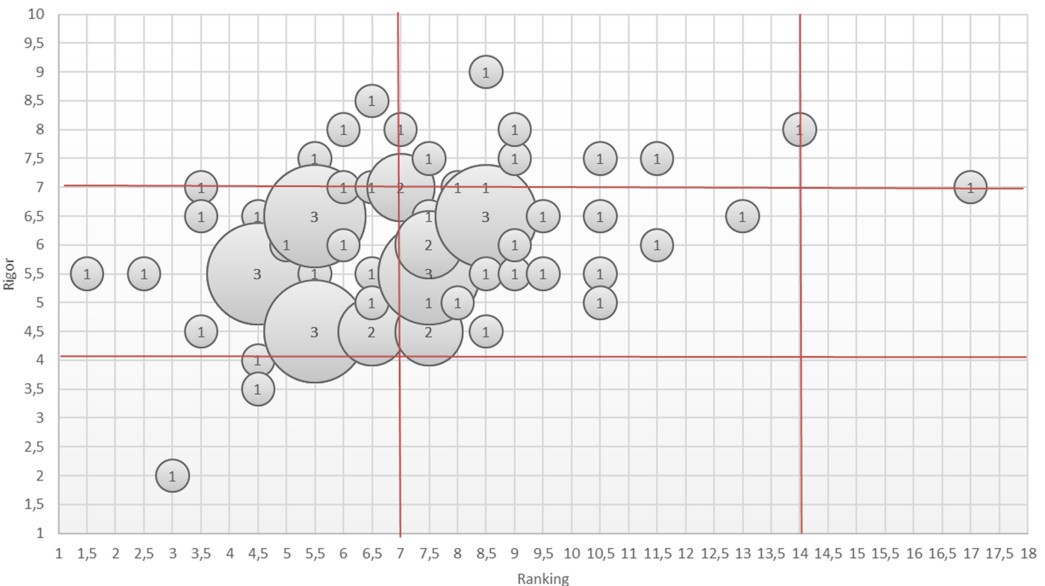

**Figure 6 Overview of rigor-ranking scores for the papers.** Overview of rigor-ranking scores for the papers.

(nine papers of 30) rigor in this period. In 2008–2011, half of the papers analyzed have a low ranking, and the other half of the papers have a middle ranking (15 papers of 22). In 2014–2016, half of the papers (11 papers of 22) have a low ranking. The remainder of the papers have a middle (nine papers of 220) or high (two papers of 22) ranking in this period.

In Fig. 6, we present the ranking of dependency based on the rigor of papers. The size of the bubbles depends on the number of papers with the same ranking-rigor score. The red lines represent intervals of rigor and ranking, as described previously.

The number of papers according to the rigor-ranking scores are presented in Table 11.

As can be seen from Table 11, the largest portion of the papers is distributed in the interval of middle to high rigor and middle to high ranking (63 papers of 67). The remaining other sets are much smaller (see Table 11 and Fig. 6).

In Fig. 7, we have presented the summed dependency of rankings according to the papers' publication years. The size of the bubbles depends on the rigor of the papers.

**Table 11 The number of papers according to the rigor-ranking scores.**

| Ranking rigor | Low ranking [1; 7) | Middle ranking [7; 14) | High ranking [14; 22] |
|---|---|---|---|
| Low rigor [2; 4) | 2 | 0 | 0 |
| Middle rigor [4; 6) | 15 | 13 | 0 |
| High rigor [6; 9] | 13 | 22 | 2 |

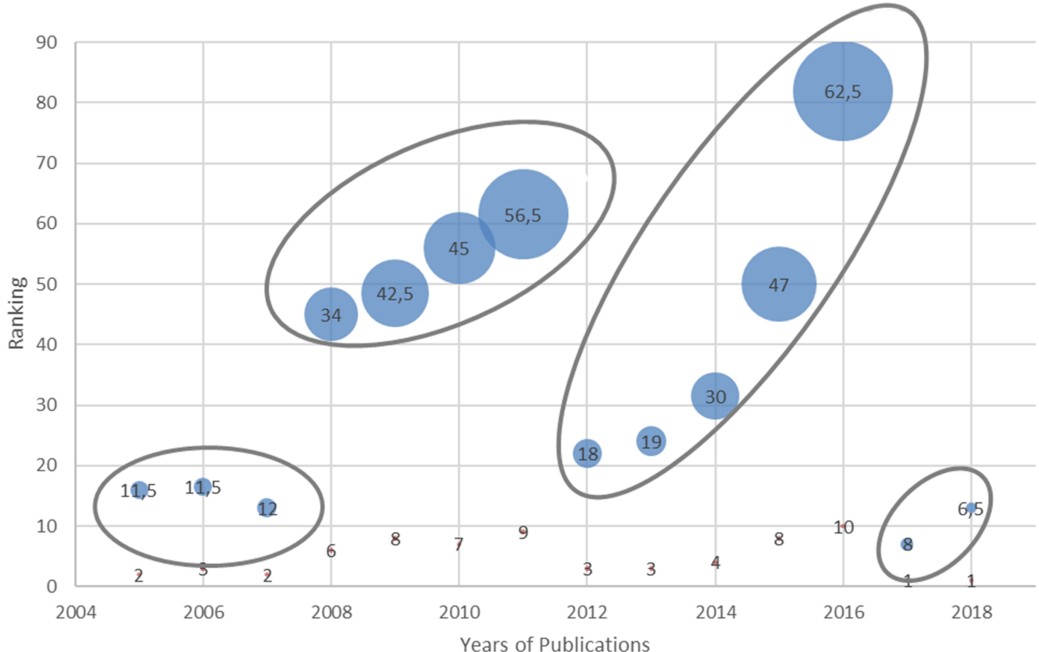

**Figure 7 Ranking of papers according to years.** Ranking of papers according to years.

Ranking and rigor is summarized by the years, and from this figure we can see that all studies can be divided into four sets as listed below:

1. 1. Set "A" (the earlier period of 2005–2007)–papers of this period have low or middle rigor [2; 6] and low or middle ranking [7; 14].
2. 2. Set "B" (the middle period of 2008–2011)–papers of this period have above the middle rigor [4; 6] and above the middle ranking [7; 14].
3. 3. Set "C + D" (the later period of 2012–2018)–papers of this period have different rigor and ranking, starting from low values and finishing at high values.

In Fig. 7, a sequence of numbers (red bubbles) shows the number of papers per year. The numbers of papers according to the rigor-ranking scores in the periods identified are presented in Table 12.

As can be seen from Table 12 and Fig. 7, the tendency of distribution of papers according to the rigor-ranking scores in the periods identified remained the same. The

**Table 12 Numbers of papers according to the rigor-ranking scores in identified periods.**

| Ranking rigor | Low ranking [1; 7) (A/B/C) | Middle ranking [7; 14) (A/B/C) | High ranking [14; 22) (A/B/C) |
|---|---|---|---|
| Low rigor [2; 4) (A/B/C) | 1/1/0 | 0/0/0 | 0/0/0 |
| Middle rigor [4; 6) (A/B/C) | 4/11/0 | 0/4/9 | 0/0/0 |
| High rigor [6; 9] (A/B/C) | 2/11/0 | 0/3/19 | 0/0/2 |

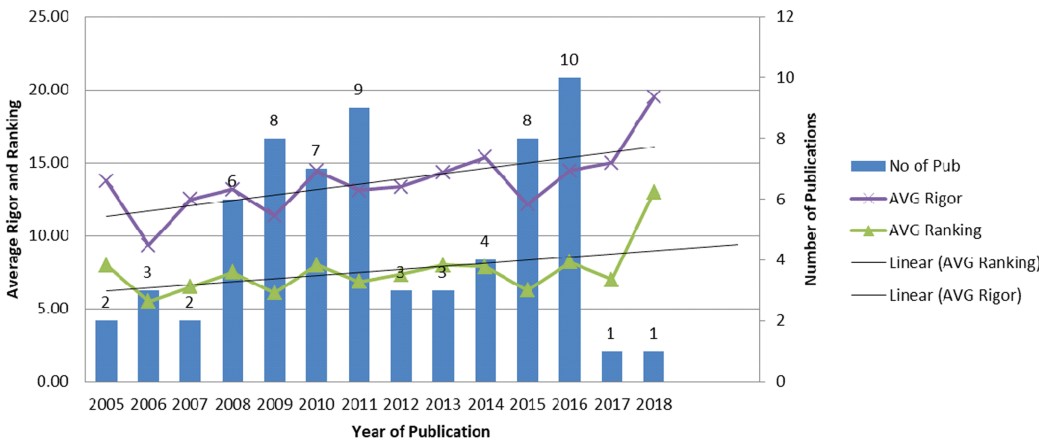

**Figure 8 The evolution of research on BP dynamicity.**

largest portion of papers have a high rigor and a middle ranking. Another large, but slightly smaller, set of papers has a middle rigor and a low or middle ranking. Other sets are generally much smaller (see Table 12 and Fig. 7).

To analyze the evolution of research, Fig. 8 shows how the variables of rigor and relevance have changed over time together with the number of papers included each year. Rigor and relevance are scored on the left y-axis, while the number of papers is given on the right y-axis. Rigor is in a range of [3.83–8.00]. Ranking is in a range of [5.50–13.00]. The results indicate an improvement in both rigor and ranking (see the linear trend line for both AVG Ranking and AVG Rigor).

## Implementation aspects of DBP

As our primary research, as presented in *Vasilecas, Kalibatiene & Lavbič (2016)* was on DBP modeling and simulation approaches and tools, in this section we review the implementation architectures proposed in the papers analyzed.

More detailed analysis of the implementation perspectives of the selected papers was conducted because of the practical interest. The authors (*Hermosillo, Seinturier & Duchien, 2010*; *Hermosillo, 2012*) propose the CEVICHE framework and deal with the architecture for DBP adaptation. The main drawback of the suggested approach is that decision or variation point and adaptation rules should be predefined in the BP model and CEVICHE can only handle known specific cases (*Pourshahid et al., 2012*). Thus, the proposed framework is not general and can be used to execute already known alternative processes

after detecting predefined situations. *van der Aalst, Weske & Grünbauer (2005)* identified and classified the main entities of case handling systems in a meta-model and implemented it in the case-handling system FLOWer using a realistic example. Moreover, as the authors (*van der Aalst, Weske & Grünbauer, 2005*) stated, more flexibility may pose many problems, ranging from unauthorized actions to incomplete cases. *van der Aalst, Pesic & Schonenberg (2009)* state that the main limitations of their proposed declarative approach are as follows: (a) a constraint-based approach is not very suitable for processes that are of a strict procedural nature; (b) declarative workflow specifications may be less readable if many (interacting) constraints are added; and (c) the efficiency of the Declare engine decreases when dealing with large specifications.

Among existing PAIS, there are highlighted: YAWL (*van der Aalst & Ter Hofstede, 2005*), Flower (*Dumas, van der Aalst & Ter Hofstede, 2005*), DECLARE (*Pesic & van der Aalst, 2006*, September) and ADEPT (*Reichert, Rinderle & Dadam, 2003*), where each supports flexibility in a different way. *Santo Carvalho, Santoro & Revoredo (2015)* state that none of them support adaptation in terms of adequacy for business, but only in ensuring that any adjustment will not somehow corrupt the process. Moreover, the analysis and consequent changes proposed in these PAIS are done manually. A decision on adaptation is not recorded, nor is an analysis made regarding it.

In *Hallerbach, Bauer & Reichert (2010)*, a number of Provop process variants are defined and proposed through a set of change operations in Provop, which forces the deployment of all process variants using conditional branching. According to *Ayora et al. (2012)*, September; 2012), this is not an example of real adaptability, as all alternatives are transformed into executable versions.

*Jiang et al. (2016)* found that expressing dynamicity through Constraint-based declarative approaches are not good at precisely describing the procedure of process models, tending to describe all the activities of the process model and the main constraints describing the relationships among these activities, as DECLARE (*Pesic, Schonenberg & van der Aalst, 2007*) and ConDec (*Pesic & van der Aalst, 2006*). They therefore have to work together with imperative approaches.

An approach presented in *Jiang et al. (2016)* is based on vertical and horizontal activity refinement. However, as the authors state, the dynamic refinement of flexible activities remains a challenge in the modeling and application of flexible workflows. *Jiang, Li & Yang (2017)* propose the use of a knowledge tree and various constraints to solve this issue.

As the analysis shows, the most significant part of these approaches is implemented as a case study (as in (*Mejia Bernal et al., 2010*; *Hermosillo, Seinturier & Duchien, 2010*), or the automation of the proposed approach at some level.

## Analysis of the existing tools on DBP modeling and simulation

For more comprehensive study, some BP modeling and simulation tools are compared in Table 13 according to the previously defined requirements (DBPR). Table 13 shows whether a system provides full (+), partial (+/−) or no support (−) for a particular feature (see DBPR).

**Table 13 Comparison of DBP modeling and simulation tools.**

| Comparison criteria | IBM Websphere[1] (v.7.0 2014) | Simprocess[2] (v 2015) | Simul8[3] | AccuProcess[4] | ARIS 9.7[5] |
|---|---|---|---|---|---|
| 1. Process model predefined (DBPR-1) | | | | | |
| 1.1. Predefined process model | + | + | + | + | + |
| 2. Context predefined (DBPR-2) | | | | | |
| 2.1 Internal context (or resource model) could be defined before simulation/execution | + | + | + | + | + |
| 2.2 External context could be defined before simulation/execution | – | – | – | – | – |
| 3. Rules predefined (DBPR-3) | | | | | |
| 3.1. Rules integration into BP | + | + | + | + | + |
| 3.2. Rules management | + | + | + | + | + |
| 3.3. Comparison of rules | + | – | – | – | +/- |
| 3.4. Analysis of rules | + | – | – | – | – |
| 4. Accumulation of experience (DBPR-5) | | | | | |
| 4.1. Usage of historical data | + | + | + | + | + |
| 4.2. Analysis of historical data | +/– | +/– | +/– | – | + |
| 5. Changes at runtime (DBPR-4) | | | | | |
| 5.1. Changes of rules | – | – | – | – | – |
| 5.2. Changes of context | – | – | – | – | – |
| 5.3. Changes of activities | – | – | +/– | + | +/– |

**Notes:**
1 http://www-03.ibm.com/software/products/en/modeler-advanced.
2 http://simprocess.com.
3 http://www.simul8.com.
4 https://www.bpmleader.com/accuprocess-accuprocess-modeler/.
5 http://www.ariscommunity.com/business-process-simulation.

We can determine from Table 13 that all of the five tools analyzed need an initial BP model (1). This limits the dynamicity of BP. All BP modeling and simulation tools allow the allocation of resources to activities (2.1). In some analyzed tools, like IBM Websphere and Simprocess, it is possible to define not only the cost and time attributes of the resource, but also other attributes such as volume, color, etc. It is not possible to define external context (2.2) in all five analyzed tools. Whilst all of the tools support features of rule modeling within BP; none of them support an analysis of these rules at BP runtime (3.4). Usage of historical data (4) is supported in almost all of the tools analyzed (except AssuProcess). As can be seen from Table 13, existing tools are well developed to model, analyze, and simulate static BP (5). However, those tools are not suitable for the proper simulation/execution of DBP.

## Discussion, answers to the research questions and future work

In this section, we present the discussion regarding the results obtained on BP dynamicity, and therefore the answers to the research questions posed in this paper.

The most frequently proposed approaches on BP dynamicity deal with the predefined process model, in which some dynamicity is ensured through changing a process model or a process instance at the design stage or at run-time by adding or restricting some

behavior. However, adaptation or customization of a process based on a predefined process model allows the achievement of only some level of dynamicity (RQ1). Some authors propose the generation of a process model or its instance from a set of activities or process fragments. These fragments can vary according to the contexts and rules at process instance run-time. In many cases, however, activities or fragments, variants, and rules are predefined that limit process dynamicity (RQ1). In some domains, it is difficult or impossible to predict all possible changes of a context and thus predefine possible sequences of activities, fragments, or rules. Therefore, the most dynamic BP should not have a predefined set and sequence of activities. Instead, a goal should be known and defined. One way to achieve the goal is to apply a multi criteria process optimization. A goal-oriented approach implies a focus on achieving some state in the system's resources, instead of the execution of a predefined set and sequence of activities. The primary advantage of this approach is that we achieve more dynamicity and the strong goal-orientation of a BP (RQ1).

According to *Cangemi & Taylor (2018)*, artificial intelligence (AI) can provide support in choosing the next most suitable activity at process instance run-time, i.e., answering the question of which activity should be executed. As processes become increasingly branched and individualized early, AI can help to route the respective flow units through the process (*Satyal et al., 2018*, October). Additionally, AI-algorithms can determine patterns in the data and create new process variants, as the process is being executed in the shadows, without affecting customers or process actors (*Satyal et al., 2018*, October). We offer the dynamicity of BP obtained through BRs, as presented in *Vasilecas, Kalibatiene & Lavbič (2016)*. Moreover, in this research the topic of AI is out of scope.

During the analysis of BP dynamicity in the papers selected, the difference of approaches lies largely at the process instance level. These differences are as follows: whether the process instance can be changed at run-time and to what extent; whether the process instance responds to changes in the internal and external contexts; whether there are any rules for choosing the next activity for the execution; whether the process instance is executed according to the process goal; and whether the process instance uses the experience gained previously (RQ2).

We can now define the main features of a dynamic BP (RQ3) (see Table 14), which are determined based on this SLR. As can be seen from the obtained results, the most relevant features are the following: rule-based, changes at run-time, context-sensitive, and DBP feasibility. The importance of the rule-based feature can be explained by the phenomenon that BRs are widely agreed between researchers and practitioners as a way to express business knowledge and govern BP (*Valente et al., 2016a*, *2016b*; *Vasilecas, Kalibatiene & Lavbič, 2016*). Consequently, there is a number of researches appling busyiness rules for ensuring BP dynamicity. The changes at run-time feature reflect the business's need to adapt to changing environmental conditions continually. The importance of the context-sensitive feature means that for the successful BP it should react not only to the changing environment, but also to the needs of the customer. Finally, DBP should be implemented in a particular way to support its business effectively.

**Table 14 Main features of a dynamic BP (RQ3).**

| Feature | Explanation | Origin | |
|---|---|---|---|
| Not predefined process model | The set of process activities is defined partially, and additional activities can be defined at run-time, if needed. The sequence of process activities is not predefined before process execution, and every next activity is chosen to fit a BP goal. | DBPR-1<br>Table 4 feature 1<br>Table 8, the first three columns (1.1, 1.2, and 1.3) | Found in 8 of 67 papers |
| Context-sensitive | Process instance should respond to the changing context, which is external or internal. | DBPR-2<br>Table 4 feature 2<br>Table 8, columns 2.1, 2.2, and 2.3 | Found in 50 of 67 papers |
| Rule-based | The sequence of process activities and content of each activity depend on rules, which are defined, and new rules can be added, if needed, at run-time. | DBPR-3<br>Table 4 feature 3<br>Table 8, columns 3.1–3.4 | Found in 55 of 67 papers |
| Changes at run-time | All possible changes can be done by any role involved at process instance run-time with possibly low latency. | DBPR-4<br>Table 4 feature 4<br>Table 8, columns 4, 1.3, 2.3, 3.2 | Found in 55 of 67 papers |
| Goal-oriented | The selection of each process activity should be aligned with a process goal. | DBPR-6<br>Table 4 feature 6<br>Table 8, column 6 | Found in 19 of 67 papers |
| Aligned with a process ontology | All process components should be aligned with a process ontology. | Table 4 feature 7<br>Table 8, column 7 | Found in 10 of 67 papers |
| Accumulation of experience | The selection of each process activity should be aligned with historical data, accumulated throughout this process. | DBPR-5<br>Table 4 feature 5<br>Table 8, columns 5.1–5.5 | Found in 12 of 67 papers |
| Feasibility | The feasibility of the process should be ensured. | Table 4 features 9 and 10<br>Table 8, columns 9.1–9.5, 10.1 – 10.5 | Found in 63 of 67 papers |

Other found DBP features (i.e., goal-oriented, accumulation of experience, align with a process ontology and not predefined process model) are not so common with comparison to the previously discussed features. This is because those features are not directly expressed in a process model. BPs are designed and implemented into software systems to achieve business goals. However, many existing classic BP modeling approaches do not provide means for the direct expression of a business goal. Most often, the goal is expressed in BP by BRs. The same is with ontology, i.e., it is usually meant but not directly expressed and used for BP modeling.

Summing up, the automation of nowadays BP cannot be achieved by solely focusing on process activities and using classical BP modeling techniques. It requires a holistic approach that also captures the business environment's social aspects, such as corporate strategies, organization policies, negotiations, and cooperation (*Kir & Erdogan, 2021*).

The results obtained highlight the main limitations and gaps related to dynamic BP (RQ4). In the research area, surrounding this topic there is a lack of research and approaches that consider how dynamic BP achieves a goal and how this is implemented. The advantages of the mining and usage of historical data is well known in different areas. However, there are no approaches relating to the accumulation of experience during

dynamic process execution and subsequent use of it for more effective goal implementation. A limited amount of research and approaches enable the rule changing during process instance execution, which is especially important when working with long-running processes.

An in-depth analysis of the selected papers, according to their perspectives on implementation, shows that the implementation and collection of experimental results remains insufficient in most research projects. Therefore, DBP modeling and implementation still requires more research.

The analysis of the selected literature reviews on BP dynamicity shows that they are not analyzing dynamicity as a feature of a BP (RQ5). They analyze process variability, but not enough attention is placed on BP dynamicity. Gaps in literature reviews still remain, primarily in presenting a research methodology, considering threats to validity, and the quality of assessment of the search performed.

Moreover, the analysis of the visualized results allows us to state that a portion of these papers lack a clear description of the approaches presented. The largest portion of these papers have no description of the proposed approach, lack models on DBP in a formal language, and have no evaluation of the results obtained. At first glance, the majority of papers have some implementation of the proposed approach by presenting a case study, simulation examples, or a prototype. However, these are not complete, and perhaps the lack of an evaluation of the results obtained can be explained by the complexity of this task (RQ6).

## CONCLUSIONS

In this paper, we have presented a literature review on DBP. After applying a research procedure based on keywords and inclusion and exclusion quality criteria, we identified 67 papers for detailed study. Those papers were assessed based on the features defined, which allowed us to determine a scientific ranking, denoting the level of dynamicity of the presented approach, and rigor. We were also able to denote the level of description of the approach presented in each paper, and to answer the research questions initially defined in this paper.

The processed and visualized results allow us to determine that the approaches presented reach the middle level of dynamicity. We are also able to note that topics remain for further development, such as the presence or absence of a more dynamic initial process model, the usage of historic data for BP dynamicity including quality cost calculation, goal-orientation in DBP, and the semantics of DBP.

The analysis of the selected approaches allows us to identify the main features of DBP as follows: an undefined process model; process instances that respond to the changing context; the sequence of process activities and content of each activity being dependent on rules; all possible changes being able to be made by any role involved at process instance run-time with possibly low latency; the selection of each process activity being aligned with a process goal; all process components being aligned with a process ontology; the selection of each process activity being aligned with historical data; and the feasibility of the process being ensured.

According to the analysis of literature reviews on DBP, they tend to analyze the variability of processes, but not enough attention is given to BP dynamicity. The main issues in literature reviews remain in presenting a research methodology, threats to validity, and the quality assessment of the search being performed.

The following future directions are defined:

- in depth analysis of modeling languages and standards that cover DBP.
- an interdependence analysis among the defined research questions.
- in depth analysis of context in the definition of DBP (i.e., how authors define and model these contexts? What is the relationship between ontologies and context models?).
- in depth analysis of technical details about modeling and implementing of DBP.
- extending the current study by including the newest papers on DBP.
- performing a statistical analysis of the obtained results.

# APPENDIX 1

| Table 15 Categorization of references used to answer the main research questions on BP dynamicity. | | |
|---|---|---|
| **RQ** | **Process dynamicity influencing features** | **References** |
| RQ1 RQ2 RQ3 RQ4 | 1. Process model predefined (Essential) (see DBPR-1): | |
| | 1.1. Does a process have a predefined sequence of activities, i.e., a predefined process model? | (*Boffoli, Cimitile & Maggi, 2009*); (*Cao et al., 2009*); (*Dadam & Reichert, 2009*); (*Asuncion, Iacob & van Sinderen, 2010*); (*Châtel, Malenfant & Truck, 2010*); (*Bucchiarone et al., 2011*); (*Ayora et al., 2012*); (*Boffoli et al., 2012*); (*Bucchiarone, Mezzina & Pistore, 2013*); (*Abderrahmane, Mili & Boubaker, 2014*); (*Borrego & Barba, 2014*); (*Awadid & Gnannouchi, 2015*); (*Cherif, Djemaa & Amous, 2015*); (*Deb, Chaki & Ghose, 2015*); (*Ayora et al., 2016*); (*Bögl, Natschläger & Geist, 2016*); (*Czepa et al., 2016*); (*Demeyer et al., 2010*); (*Gong & Janssen, 2012*); (*Haarmann et al., 2015*); (*Hallerbach, Bauer & Reichert, 2010*); (*Hermosillo, Seinturier & Duchien, 2010*); (*Jiang et al., 2016*); (*Kannengiesser et al., 2014*); (*Kapuruge, Han & Colman, 2011*); (*Kim, Kim & Kim, 2007*); (*Li & Du, 2016*); (*Marconi et al., 2009*); (*Martinho, Domingos & Varajão, 2015*); (*Mattos et al., 2014*); (*Mejia Bernal et al., 2010*); (*Mounira & Mahmoud, 2010*); (*Natschläger et al., 2016*); (*Oberhauser, 2016*); (*Prasad et al., 2015*); (*Ramakrishnan, 2009*); (*Saidani & Nurcan, 2006*); (*Santo Carvalho, Santoro & Revoredo, 2015*); (*Santo Carvalho et al., 2013*); (*Santos et al., 2013*); (*Santos et al., 2011*); (*Sarno et al., 2015*); (*Sprovieri et al., 2016*); (*Sun, Huang & Meng, 2011*); (*Ukor & Carpenter, 2009*); (*van Eijndhoven, Iacob & Ponisio, 2008*); (*Weidlich et al., 2011*); (*Xia & Wei, 2008*); (*Yoo et al., 2008*); (*Yousfi, Saidi & Dey, 2016*); (*Yuliang et al., 2009*); (*Gottschalk et al., 2008*); (*Maggi et al., 2012*); (*Pesic & van der Aalst, 2006*); (*Wang & Wang, 2006*); (*Baresi & Guinea, 2005*); (*van der Aalst, Pesic & Schonenberg, 2009*); (*Milanovic, Gasevic & Rocha, 2011*);(*Bucchiarone et al., 2017*) |
| | 1.2. Is a process based on meta-model, general model, generic model, or abstract model? | (*Abderrahmane, Mili & Boubaker, 2014*); (*van der Aalst, Weske & Grünbauer, 2005*); (*Boffoli et al., 2012*); (*Boffoli, Cimitile & Maggi, 2009*); (*Bögl, Natschläger & Geist, 2016*); (*Bucchiarone et al., 2011*); (*Dadam & Reichert, 2009*); (*Haarmann et al., 2015*); (*Hallerbach, Bauer & Reichert, 2010*); (*Jiang et al., 2016*); (*Kannengiesser et al., 2014*); (*Kapuruge, Han & Colman, 2011*); (*Li & Du, 2016*); (*Marconi et al., 2009*); (*Martinho, Domingos & Varajão, 2015*); (*Mattos et al., 2014*); (*Oberhauser, 2016*); (*Prasad et al., 2015*); (*Saidani & Nurcan, 2006*); (*Santos et al., 2013*); (*Sarno et al., 2015*); (*Sprovieri et al., 2016*); (*van Eijndhoven, Iacob & Ponisio, 2008*); (*Wang & Capretz, 2007*); (*Weidlich et al., 2011*); (*Xiao et al., 2011*); (*Yoo et al., 2008*); (*Yousfi, Saidi & Dey, 2016*); (*Yuliang et al., 2009*); (*Gottschalk et al., 2008*); (*Maggi et al., 2012*); (*Pesic & van der Aalst, 2006*); (*Wang & Wang, 2006*); (*Baresi & Guinea, 2005*); (*van der Aalst, Pesic & Schonenberg, 2009*); (*Milanovic, Gasevic & Rocha, 2011*); (*Bucchiarone et al., 2017*) |
| | 1.3. Is it possible to define additional activities and to change or delete existing activities or skip to other activities at process instance run-time? | (*Abderrahmane, Mili & Boubaker, 2014*); (*van der Aalst, Weske & Grünbauer, 2005*); (*Awadid & Gnannouchi, 2015*); (*Ayora et al., 2016*); (*Ayora et al., 2012*); (*Bucchiarone et al., 2011*); (*Cao et al., 2009*); (*Czepa et al., 2016*); (*Dadam & Reichert, 2009*); (*Haarmann et al., 2015*); (*Hermosillo, Seinturier & Duchien, 2010*); (*Jiang et al., 2016*); (*Kapuruge, Han & Colman, 2011*); (*Khriss et al., 2008*); (*Kim, Kim & Kim, 2007*); (*Liu et al., 2012*); (*Marconi et al., 2009*); (*Mattos et al., 2014*); (*Mejia Bernal et al., 2010*); (*Mounira & Mahmoud, 2010*); (*Oberhauser, 2016*); (*Santos et al., 2013*); (*Sprovieri et al., 2016*); (*Gottschalk et al., 2008*); (*Maggi et al., 2012*); (*Pesic & van der Aalst, 2006*); (*Wang & Wang, 2006*); (*Baresi & Guinea, 2005*); (*van der Aalst, Pesic & Schonenberg, 2009*); (*Savickas & Vasilecas, 2018*); (*Milanovic, Gasevic & Rocha, 2011*); (*Bucchiarone et al., 2017*); (*Vasilecas, Kalibatiene & Lavbič, 2016*) |
| RQ1 RQ2 RQ3 RQ4 | 2. Context aware (Essential) (see DBPR-2): | |
| | 2.1. Does an external context affect the execution of the process? | (*Abderrahmane, Mili & Boubaker, 2014*); (*van der Aalst, Weske & Grünbauer, 2005*); (*Ayora et al., 2016*); (*Boffoli et al., 2012*); (*Boffoli, Cimitile & Maggi, 2009*); (*Bögl, Natschläger & Geist, 2016*); (*Bucchiarone, Mezzina & Pistore, 2013*); (*Bucchiarone et al., 2011*); (*Cherif, Djemaa & Amous, 2015*); (*Deb, Chaki & Ghose, 2015*); (*Demeyer et al., 2010*); (*Hallerbach, Bauer & Reichert, 2010*); (*Hermosillo, Seinturier & Duchien, 2010*); (*Jiang et al., 2016*); (*Marconi et al., 2009*); (*Mattos et al., 2014*); (*Mejia Bernal et al., 2010*); (*Mounira & Mahmoud, 2010*); (*Rong, Liu & Liang, 2008*); (*Santo Carvalho, Santoro & Revoredo, 2015*); (*Santo Carvalho et al., 2013*); (*Santos et al., 2013*); (*Santos et al., 2011*); (*Sprovieri et al., 2016*); (*Xia & Wei, 2008*); (*Xiao et al., 2011*); (*Yoo et al., 2008*); (*Yousfi, Saidi & Dey, 2016*); (*Wang & Wang, 2006*); (*Baresi & Guinea, 2005*); (*Savickas & Vasilecas, 2018*); (*Milanovic, Gasevic & Rocha, 2011*); (*Bucchiarone et al., 2017*); (*Vasilecas, Kalibatiene & Lavbič, 2016*) |

| Table 15 (continued) | | |
|---|---|---|
| RQ | Process dynamicity influencing features | References |
| | 2.2. Does an internal context affect the execution of the process? | (*Abderrahmane, Mili & Boubaker, 2014*); (*van der Aalst, Weske & Grünbauer, 2005*); (*Ayora et al., 2012*); (*Boffoli et al., 2012*); (*Bögl, Natschläger & Geist, 2016*); (*Cao et al., 2009*); (*Châtel, Malenfant & Truck, 2010*); (*Czepa et al., 2016*); (*Dadam & Reichert, 2009*); (*Khriss et al., 2008*); (*Kim, Kim & Kim, 2007*); (*Liu et al., 2012*); (*Marconi et al., 2009*); (*Mejia Bernal et al., 2010*); (*Mounira & Mahmoud, 2010*); (*Natschläger et al., 2016*); (*Oberhauser, 2016*); (*Prasad et al., 2015*); (*Ramakrishnan, 2009*); (*Rong, Liu & Liang, 2008*); (*Saidani & Nurcan, 2006*); (*Santo Carvalho et al., 2013*); (*Sprovieri et al., 2016*); (*Sun, Huang & Meng, 2011*); (*Ukor & Carpenter, 2009*); (*van Eijndhoven, Iacob & Ponisio, 2008*); (*Xia & Wei, 2008*); (*Xiao et al., 2011*); (*Yoo et al., 2008*); (*Wang & Wang, 2006*); (*Baresi & Guinea, 2005*); (*Savickas & Vasilecas, 2018*); (*Bucchiarone et al., 2017*); (*Vasilecas, Kalibatiene & Lavbič, 2016*) |
| | 2.3. Is a process instance affected by the change of a context at run-time? | (*van der Aalst, Weske & Grünbauer, 2005*); (*Ayora et al., 2016*); (*Ayora et al., 2012*); (*Bögl, Natschläger & Geist, 2016*); (*Bucchiarone, Mezzina & Pistore, 2013*); (*Bucchiarone et al., 2011*); (*Cao et al., 2009*); (*Cherif, Djemaa & Amous, 2015*); (*Czepa et al., 2016*); (*Dadam & Reichert, 2009*); (*Demeyer et al., 2010*); (*Hallerbach, Bauer & Reichert, 2010*); (*Hermosillo, Seinturier & Duchien, 2010*); (*Jiang et al., 2016*); (*Khriss et al., 2008*); (*Kim, Kim & Kim, 2007*); (*Liu et al., 2012*); (*Marconi et al., 2009*); (*Mattos et al., 2014*); (*Mejia Bernal et al., 2010*); (*Mounira & Mahmoud, 2010*); (*Oberhauser, 2016*); (*Rong, Liu & Liang, 2008*); (*Santo Carvalho, Santoro & Revoredo, 2015*); (*Santos et al., 2011*); (*Sprovieri et al., 2016*); (*Sun, Huang & Meng, 2011*); (*Ukor & Carpenter, 2009*); (*van Eijndhoven, Iacob & Ponisio, 2008*); (*Xia & Wei, 2008*); (*Xiao et al., 2011*); (*Yoo et al., 2008*); (*Yousfi, Saidi & Dey, 2016*); (*Wang & Wang, 2006*); (*Baresi & Guinea, 2005*); (*Savickas & Vasilecas, 2018*); (*Milanovic, Gasevic & Rocha, 2011*); (*Bucchiarone et al., 2017*); (*Vasilecas, Kalibatiene & Lavbič, 2016*) |
| RQ1 RQ2 RQ3 RQ4 | 3. Rule-based[1] (Essential) (see DBPR-3): | |
| | 3.1. Is every subsequent activity in a process selected according to the predefined rules at process instance run-time? | (*Abderrahmane, Mili & Boubaker, 2014*); (*van der Aalst, Weske & Grünbauer, 2005*); (*Asuncion, Iacob & van Sinderen, 2010*); (*Ayora et al., 2016*); (*Ayora et al., 2012*); (*Boffoli et al., 2012*); (*Boffoli, Cimitile & Maggi, 2009*); (*Bögl, Natschläger & Geist, 2016*); (*Borrego & Barba, 2014*); (*Bucchiarone, Mezzina & Pistore, 2013*); (*Bucchiarone et al., 2011*); (*Cao et al., 2009*); (*Châtel, Malenfant & Truck, 2010*); (*Czepa et al., 2016*); (*Dadam & Reichert, 2009*); (*Deb, Chaki & Ghose, 2015*); (*Demeyer et al., 2010*); (*Gong & Janssen, 2012*); (*Haarmann et al., 2015*); (*Hallerbach, Bauer & Reichert, 2010*); (*Hermosillo, Seinturier & Duchien, 2010*); (*Jiang et al., 2016*); (*Kapuruge, Han & Colman, 2011*); (*Liu et al., 2012*); (*Marconi et al., 2009*); (*Martinho, Domingos & Varajão, 2015*); (*Mattos et al., 2014*); (*Mejia Bernal et al., 2010*); (*Mounira & Mahmoud, 2010*); (*Natschläger et al., 2016*); (*Oberhauser, 2016*); (*Prasad et al., 2015*); (*Rong, Liu & Liang, 2008*); (*Saidani & Nurcan, 2006*); (*Santo Carvalho, Santoro & Revoredo, 2015*); (*Santo Carvalho et al., 2013*); (*Santos et al., 2011*); (*Sprovieri et al., 2016*); (*Sun, Huang & Meng, 2011*); (*Ukor & Carpenter, 2009*); (*van Eijndhoven, Iacob & Ponisio, 2008*); (*Wang & Capretz, 2007*); (*Weidlich et al., 2011*); (*Xia & Wei, 2008*); (*Xiao et al., 2011*); (*Yoo et al., 2008*); (*Yousfi, Saidi & Dey, 2016*); (*Yuliang et al., 2009*); (*Gottschalk et al., 2008*); (*Maggi et al., 2012*); (*Pesic & van der Aalst, 2006*); (*Wang & Wang, 2006*); (*Baresi & Guinea, 2005*); (*van der Aalst, Pesic & Schonenberg, 2009*); (*Savickas & Vasilecas, 2018*); (*Bucchiarone et al., 2017*); (*Vasilecas, Kalibatiene & Lavbič, 2016*) |
| | 3.2. Is it possible to define new rules, or change or delete existing rules, at process instance run-time? | (*Czepa et al., 2016*); (*Mejia Bernal et al., 2010*); (*Santo Carvalho, Santoro & Revoredo, 2015*); (*Yoo et al., 2008*); (*Milanovic, Gasevic & Rocha, 2011*); (*Vasilecas, Kalibatiene & Lavbič, 2016*) |
| | 3.3. Does process instance react to the rule changes made at process instance run-time? | (*Czepa et al., 2016*); (*Mejia Bernal et al., 2010*); (*Santo Carvalho, Santoro & Revoredo, 2015*); (*Yoo et al., 2008*); (*Milanovic, Gasevic & Rocha, 2011*); (*Vasilecas, Kalibatiene & Lavbič, 2016*) |
| | 3.4. Are rules used to solve other tasks? | (*Abderrahmane, Mili & Boubaker, 2014*); (*Asuncion, Iacob & van Sinderen, 2010*); (*Bögl, Natschläger & Geist, 2016*); (*Borrego & Barba, 2014*); (*Bucchiarone, Mezzina & Pistore, 2013*); (*Bucchiarone et al., 2011*); (*Châtel, Malenfant & Truck, 2010*); (*Czepa et al., 2016*); (*Dadam & Reichert, 2009*); (*Jiang et al., 2016*); (*Kannengiesser et al., 2014*); (*Kim, Kim & Kim, 2007*); (*Li & Du, 2016*); (*Martinho, Domingos & Varajão, 2015*); (*Ramakrishnan, 2009*); (*Santos et al., 2013*); (*Santos et al., 2011*); (*Weidlich et al., 2011*); (*Xia & Wei, 2008*); (*Maggi et al., 2012*) |

| Table 15 (continued) | | |
|---|---|---|
| RQ | Process dynamicity influencing features | References |
| RQ1 RQ2 RQ3 RQ4 | 4. Changes (Essential): Are changes implemented at process instance run-time with low latency? (see DBPR-4) | (*Abderrahmane, Mili & Boubaker, 2014*); (*van der Aalst, Weske & Grünbauer, 2005*); (*Awadid & Gnannouchi, 2015*); (*Ayora et al., 2016*); (*Ayora et al., 2012*); (*Bögl, Natschläger & Geist, 2016*); (*Bucchiarone, Mezzina & Pistore, 2013*); (*Bucchiarone et al., 2011*); (*Cao et al., 2009*); (*Cherif, Djemaa & Amous, 2015*); (*Czepa et al., 2016*); (*Dadam & Reichert, 2009*); (*Demeyer et al., 2010*); (*Gong & Janssen, 2012*); (*Haarmann et al., 2015*); (*Hallerbach, Bauer & Reichert, 2010*); (*Hermosillo, Seinturier & Duchien, 2010*); (*Jiang et al., 2016*); (*Kannengiesser et al., 2014*); (*Kapuruge, Han & Colman, 2011*); (*Khriss et al., 2008*); (*Kim, Kim & Kim, 2007*); (*Liu et al., 2012*); (*Marconi et al., 2009*); (*Martinho, Domingos & Varajão, 2015*); (*Mattos et al., 2014*); (*Mejia Bernal et al., 2010*); (*Mounira & Mahmoud, 2010*); (*Oberhauser, 2016*); (*Prasad et al., 2015*); (*Ramakrishnan, 2009*); (*Rong, Liu & Liang, 2008*); (*Santo Carvalho, Santoro & Revoredo, 2015*); (*Santos et al., 2013*); (*Santos et al., 2011*); (*Sprovieri et al., 2016*); (*Sun, Huang & Meng, 2011*); (*Ukor & Carpenter, 2009*); (*van Eijndhoven, Iacob & Ponisio, 2008*); (*Wang & Capretz, 2007*); (*Xia & Wei, 2008*); (*Xiao et al., 2011*); (*Yoo et al., 2008*); (*Yousfi, Saidi & Dey, 2016*); (*Yuliang et al., 2009*); (*Gottschalk et al., 2008*); (*Maggi et al., 2012*); (*Pesic & van der Aalst, 2006*); (*Wang & Wang, 2006*); (*Baresi & Guinea, 2005*); (*van der Aalst, Pesic & Schonenberg, 2009*); (*Savickas & Vasilecas, 2018*); (*Milanovic, Gasevic & Rocha, 2011*); (*Bucchiarone et al., 2017*); (*Vasilecas, Kalibatiene & Lavbič, 2016*) |
| RQ1 RQ2 RQ3 RQ4 | 5. Accumulation of experience (see DBPR-5): | |
| | 5.1. Is the experience of each process instance execution stored? | (*Bögl, Natschläger & Geist, 2016*); (*Borrego & Barba, 2014*); (*Bucchiarone et al., 2011*); (*Cao et al., 2009*); (*Haarmann et al., 2015*); (*Hermosillo, Seinturier & Duchien, 2010*); (*Mounira & Mahmoud, 2010*); (*Santo Carvalho, Santoro & Revoredo, 2015*); (*Santo Carvalho et al., 2013*); (*Xia & Wei, 2008*); (*Savickas & Vasilecas, 2018*); (*Vasilecas, Kalibatiene & Lavbič, 2016*) |
| | 5.2. Is it possible to classify executed instances of a process as a "good practice" and a "bad practice"? | (*Hermosillo, Seinturier & Duchien, 2010*); (*Vasilecas, Kalibatiene & Lavbič, 2016*) |
| | 5.3. Are time, cost, etc. values calculated and stored for each process instance? | (*Bögl, Natschläger & Geist, 2016*); (*Borrego & Barba, 2014*); (*Haarmann et al., 2015*); (*Santo Carvalho, Santoro & Revoredo, 2015*); (*Vasilecas, Kalibatiene & Lavbič, 2016*) |
| | 5.4. Is it possible to restrict the execution of process instances that are named "bad instances"? | (*Vasilecas, Kalibatiene & Lavbič, 2016*) |
| | 5.5. Is historical data used for solving other related tasks? | (*Abderrahmane, Mili & Boubaker, 2014*); (*van der Aalst, Weske & Grünbauer, 2005*); (*Bögl, Natschläger & Geist, 2016*); (*Borrego & Barba, 2014*); (*Bucchiarone et al., 2011*); (*Cao et al., 2009*); (*Haarmann et al., 2015*); (*Mattos et al., 2014*); (*Mounira & Mahmoud, 2010*); (*Santo Carvalho, Santoro & Revoredo, 2015*); (*Santo Carvalho et al., 2013*); (*Savickas & Vasilecas, 2018*) |
| RQ1 RQ2 RQ3 RQ4 | 6. Goal: is a process goal-oriented? (see DBPR-6) | (*Asuncion, Iacob & van Sinderen, 2010*); (*Ayora et al., 2012*); (*Borrego & Barba, 2014*); (*Bucchiarone, Mezzina & Pistore, 2013*); (*Bucchiarone et al., 2011*); (*Deb, Chaki & Ghose, 2015*); (*Gong & Janssen, 2012*); (*Kapuruge, Han & Colman, 2011*); (*Mejia Bernal et al., 2010*); (*Saidani & Nurcan, 2006*); (*Santo Carvalho, Santoro & Revoredo, 2015*); (*Santo Carvalho et al., 2013*); (*Sprovieri et al., 2016*); (*Wang & Capretz, 2007*); (*Xia & Wei, 2008*); (*Yousfi, Saidi & Dey, 2016*); (*Wang & Wang, 2006*); (*Savickas & Vasilecas, 2018*); (*Vasilecas, Kalibatiene & Lavbič, 2016*) |
| RQ1 RQ2 RQ3 RQ4 | 7. Ontology: is domain ontology or process ontology used in an approach? | (*Abderrahmane, Mili & Boubaker, 2014*); (*Châtel, Malenfant & Truck, 2010*); (*Czepa et al., 2016*); (*Gong & Janssen, 2012*); (*Kannengiesser et al., 2014*); (*Martinho, Domingos & Varajão, 2015*); (*Mattos et al., 2014*); (*Prasad et al., 2015*); (*Ramakrishnan, 2009*); (*Santo Carvalho, Santoro & Revoredo, 2015*) |

(Continued)

| RQ | Process dynamicity influencing features | References |
|---|---|---|
| RQ2 RQ3 RQ4 RQ6 | **8. Approach description:** | |
| | 8.1. Does the paper present a clear description of the proposed approach on BP dynamicity? | (*Abderrahmane, Mili & Boubaker, 2014*); (*van der Aalst, Weske & Grünbauer, 2005*); (*Asuncion, Iacob & van Sinderen, 2010*); (*Awadid & Gnannouchi, 2015*); (*Ayora et al., 2016*); (*Ayora et al., 2012*); (*Boffoli et al., 2012*); (*Boffoli, Cimitile & Maggi, 2009*); (*Bögl, Natschläger & Geist, 2016*); (*Borrego & Barba, 2014*); (*Bucchiarone, Mezzina & Pistore, 2013*); (*Bucchiarone et al., 2011*); (*Cao et al., 2009*); (*Châtel, Malenfant & Truck, 2010*); (*Cherif, Djemaa & Amous, 2015*); (*Czepa et al., 2016*); (*Dadam & Reichert, 2009*); (*Deb, Chaki & Ghose, 2015*); (*Demeyer et al., 2010*); (*Gong & Janssen, 2012*); (*Haarmann et al., 2015*); (*Hallerbach, Bauer & Reichert, 2010*); (*Hermosillo, Seinturier & Duchien, 2010*); (*Jiang et al., 2016*); (*Kannengiesser et al., 2014*); (*Kapuruge, Han & Colman, 2011*); (*Khriss et al., 2008*); (*Kim, Kim & Kim, 2007*); (*Li & Du, 2016*); (*Liu et al., 2012*); (*Marconi et al., 2009*); (*Martinho, Domingos & Varajão, 2015*); (*Mattos et al., 2014*); (*Mejia Bernal et al., 2010*); (*Mounira & Mahmoud, 2010*); (*Natschläger et al., 2016*); (*Oberhauser, 2016*); (*Prasad et al., 2015*); (*Ramakrishnan, 2009*); (*Rong, Liu & Liang, 2008*); (*Saidani & Nurcan, 2006*); (*Santo Carvalho, Santoro & Revoredo, 2015*); (*Santo Carvalho et al., 2013*); (*Santos et al., 2013*); (*Santos et al., 2011*); (*Sarno et al., 2015*); (*Sprovieri et al., 2016*); (*Sun, Huang & Meng, 2011*); (*Ukor & Carpenter, 2009*); (*van Eijndhoven, Iacob & Ponisio, 2008*); (*Wang & Capretz, 2007*); (*Weidlich et al., 2011*); (*Xia & Wei, 2008*); (*Xiao et al., 2011*); (*Yoo et al., 2008*); (*Yousfi, Saidi & Dey, 2016*); (*Yuliang et al., 2009*); (*Gottschalk et al., 2008*); (*Maggi et al., 2012*); (*Pesic & van der Aalst, 2006*); (*Wang & Wang, 2006*); (*Baresi & Guinea, 2005*); (*van der Aalst, Pesic & Schonenberg, 2009*); (*Savickas & Vasilecas, 2018*); (*Milanovic, Gasevic & Rocha, 2011*); (*Bucchiarone et al., 2017*); (*Vasilecas, Kalibatiene & Lavbič, 2016*) |
| | 8.2. Does the paper present a description of the proposed approach on dynamic BP in a formal language? | (*van der Aalst, Weske & Grünbauer, 2005*); (*Borrego & Barba, 2014*); (*Bucchiarone, Mezzina & Pistore, 2013*); (*Deb, Chaki & Ghose, 2015*); (*Demeyer et al., 2010*); (*Hallerbach, Bauer & Reichert, 2010*); (*Hermosillo, Seinturier & Duchien, 2010*); (*Jiang et al., 2016*); (*Kannengiesser et al., 2014*); (*Kapuruge, Han & Colman, 2011*); (*Li & Du, 2016*); (*Liu et al., 2012*); (*Marconi et al., 2009*); (*Martinho, Domingos & Varajão, 2015*); (*Mattos et al., 2014*); (*Santo Carvalho, Santoro & Revoredo, 2015*); (*Santos et al., 2013*); (*Sun, Huang & Meng, 2011*); (*Ukor & Carpenter, 2009*); (*Wang & Capretz, 2007*); (*Weidlich et al., 2011*); (*Xia & Wei, 2008*); (*Xiao et al., 2011*); (*Yoo et al., 2008*); (*Yousfi, Saidi & Dey, 2016*); (*Maggi et al., 2012*); (*Wang & Wang, 2006*); (*Savickas & Vasilecas, 2018*); (*Bucchiarone et al., 2017*) |
| RQ1 RQ2 RQ3 RQ4 RQ6 | **9. Dynamicity implementation approach (if 1.1. is yes):** | |
| | 9.1. Is the process dynamicity realized through variants? | (*Abderrahmane, Mili & Boubaker, 2014*); (*Ayora et al., 2016*); (*Ayora et al., 2012*); (*Boffoli et al., 2012*); (*Boffoli, Cimitile & Maggi, 2009*); (*Bögl, Natschläger & Geist, 2016*); (*Bucchiarone, Mezzina & Pistore, 2013*); (*Châtel, Malenfant & Truck, 2010*); (*Cherif, Djemaa & Amous, 2015*); (*Dadam & Reichert, 2009*); (*Hallerbach, Bauer & Reichert, 2010*); (*Hermosillo, Seinturier & Duchien, 2010*); (*Jiang et al., 2016*); (*Kim, Kim & Kim, 2007*); (*Li & Du, 2016*); (*Marconi et al., 2009*); (*Martinho, Domingos & Varajão, 2015*); (*Natschläger et al., 2016*); (*Oberhauser, 2016*); (*Ramakrishnan, 2009*); (*Santos et al., 2011*); (*Sarno et al., 2015*); (*Ukor & Carpenter, 2009*); (*van Eijndhoven, Iacob & Ponisio, 2008*); (*Wang & Capretz, 2007*); (*Weidlich et al., 2011*); (*Xia & Wei, 2008*); (*Yousfi, Saidi & Dey, 2016*); (*Gottschalk et al., 2008*); (*Milanovic, Gasevic & Rocha, 2011*); (*Bucchiarone et al., 2017*) |
| | 9.2. Is the process dynamicity realized through cases? | (*van der Aalst, Weske & Grünbauer, 2005*); (*Czepa et al., 2016*); (*Haarmann et al., 2015*); (*Sprovieri et al., 2016*) |
| | 9.3. Is the process dynamicity realized through a declarative approach? | (*Borrego & Barba, 2014*); (*Demeyer et al., 2010*); (*Gong & Janssen, 2012*); (*Mejia Bernal et al., 2010*); (*Santos et al., 2013*); (*Sun, Huang & Meng, 2011*); (*Xiao et al., 2011*); (*Maggi et al., 2012*); (*Pesic & van der Aalst, 2006*); (*Wang & Wang, 2006*); (*van der Aalst, Pesic & Schonenberg, 2009*) |
| | 9.4. Is the process dynamicity realized through goal-orientation? | (*Asuncion, Iacob & van Sinderen, 2010*); (*Ayora et al., 2012*); (*Bucchiarone et al., 2011*); (*Deb, Chaki & Ghose, 2015*); (*Gong & Janssen, 2012*); (*Mattos et al., 2014*); (*Prasad et al., 2015*); (*Santo Carvalho, Santoro & Revoredo, 2015*); (*Santo Carvalho et al., 2013*); (*Wang & Wang, 2006*); (*Savickas & Vasilecas, 2018*); (*Vasilecas, Kalibatiene & Lavbič, 2016*) |
| | 9.5. Is the process dynamicity realized through other approaches? | (*Awadid & Gnannouchi, 2015*); (*Cao et al., 2009*); (*Kannengiesser et al., 2014*); (*Kapuruge, Han & Colman, 2011*); (*Khriss et al., 2008*); (*Liu et al., 2012*); (*Mounira & Mahmoud, 2010*); (*Rong, Liu & Liang, 2008*); (*Yoo et al., 2008*); (*Yuliang et al., 2009*); (*Baresi & Guinea, 2005*) |

| Table 15 (continued) | | |
| --- | --- | --- |
| RQ | Process dynamicity influencing features | References |
| RQ2 RQ3 RQ4 RQ6 | 10. Implementation of the proposed approach: | |
| | 10.1. Is any case of implementation of a proposed model presented in the paper? | (*Abderrahmane, Mili & Boubaker, 2014*); (*van der Aalst, Weske & Grünbauer, 2005*); (*Asuncion, Iacob & van Sinderen, 2010*); (*Awadid & Gnannouchi, 2015*); (*Ayora et al., 2016*); (*Ayora et al., 2012*); (*Boffoli et al., 2012*); (*Boffoli, Cimitile & Maggi, 2009*); (*Bögl, Natschläger & Geist, 2016*); (*Borrego & Barba, 2014*); (*Bucchiarone, Mezzina & Pistore, 2013*); (*Bucchiarone et al., 2011*); (*Cao et al., 2009*); (*Châtel, Malenfant & Truck, 2010*); (*Cherif, Djemaa & Amous, 2015*); (*Czepa et al., 2016*); (*Dadam & Reichert, 2009*); (*Deb, Chaki & Ghose, 2015*); (*Demeyer et al., 2010*); (*Gong & Janssen, 2012*); (*Haarmann et al., 2015*); (*Hallerbach, Bauer & Reichert, 2010*); (*Hermosillo, Seinturier & Duchien, 2010*); (*Jiang et al., 2016*); (*Kannengiesser et al., 2014*); (*Kapuruge, Han & Colman, 2011*); (*Khriss et al., 2008*); (*Kim, Kim & Kim, 2007*); (*Li & Du, 2016*); (*Liu et al., 2012*); (*Marconi et al., 2009*); (*Martinho, Domingos & Varajão, 2015*); (*Mattos et al., 2014*); (*Mejia Bernal et al., 2010*); (*Mounira & Mahmoud, 2010*); (*Natschläger et al., 2016*); (*Oberhauser, 2016*); (*Prasad et al., 2015*); (*Ramakrishnan, 2009*); (*Rong, Liu & Liang, 2008*); (*Santo Carvalho, Santoro & Revoredo, 2015*); (*Santo Carvalho et al., 2013*); (*Santos et al., 2013*); (*Santos et al., 2011*); (*Sarno et al., 2015*); (*Sprovieri et al., 2016*); (*Sun, Huang & Meng, 2011*); (*van Eijndhoven, Iacob & Ponisio, 2008*); (*Wang & Capretz, 2007*); (*Weidlich et al., 2011*); (*Xia & Wei, 2008*); (*Xiao et al., 2011*); (*Yoo et al., 2008*); (*Yousfi, Saidi & Dey, 2016*); (*Yuliang et al., 2009*); (*Gottschalk et al., 2008*); (*Maggi et al., 2012*); (*Pesic & van der Aalst, 2006*); (*Wang & Wang, 2006*); (*Baresi & Guinea, 2005*); (*van der Aalst, Pesic & Schonenberg, 2009*); (*Savickas & Vasilecas, 2018*); (*Milanovic, Gasevic & Rocha, 2011*); (*Bucchiarone et al., 2017*); (*Vasilecas, Kalibatiene & Lavbič, 2016*) |
| | 10.3. Does the paper present a case study or a prototype or a descriptive example? | (*Abderrahmane, Mili & Boubaker, 2014*); (*van der Aalst, Weske & Grünbauer, 2005*); (*Asuncion, Iacob & van Sinderen, 2010*); (*Ayora et al., 2016*); (*Ayora et al., 2012*); (*Boffoli et al., 2012*); (*Boffoli, Cimitile & Maggi, 2009*); (*Bögl, Natschläger & Geist, 2016*); (*Borrego & Barba, 2014*); (*Bucchiarone, Mezzina & Pistore, 2013*); (*Bucchiarone et al., 2011*); (*Cao et al., 2009*); (*Châtel, Malenfant & Truck, 2010*); (*Cherif, Djemaa & Amous, 2015*); (*Czepa et al., 2016*); (*Dadam & Reichert, 2009*); (*Deb, Chaki & Ghose, 2015*); (*Demeyer et al., 2010*); (*Gong & Janssen, 2012*); (*Haarmann et al., 2015*); (*Hallerbach, Bauer & Reichert, 2010*); (*Hermosillo, Seinturier & Duchien, 2010*); (*Jiang et al., 2016*); (*Kannengiesser et al., 2014*); (*Kapuruge, Han & Colman, 2011*); (*Khriss et al., 2008*); (*Kim, Kim & Kim, 2007*); (*Li & Du, 2016*); (*Liu et al., 2012*); (*Marconi et al., 2009*); (*Martinho, Domingos & Varajão, 2015*); (*Mattos et al., 2014*); (*Mejia Bernal et al., 2010*); (*Mounira & Mahmoud, 2010*); (*Natschläger et al., 2016*); (*Oberhauser, 2016*); (*Prasad et al., 2015*); (*Ramakrishnan, 2009*); (*Rong, Liu & Liang, 2008*); (*Santo Carvalho, Santoro & Revoredo, 2015*); (*Santo Carvalho et al., 2013*); (*Santos et al., 2013*); (*Santos et al., 2011*); (*Sarno et al., 2015*); (*Sprovieri et al., 2016*); (*Sun, Huang & Meng, 2011*); (*van Eijndhoven, Iacob & Ponisio, 2008*); (*Wang & Capretz, 2007*); (*Weidlich et al., 2011*); (*Xia & Wei, 2008*); (*Xiao et al., 2011*); (*Yoo et al., 2008*); (*Yousfi, Saidi & Dey, 2016*); (*Yuliang et al., 2009*); (*Gottschalk et al., 2008*); (*Maggi et al., 2012*); (*Wang & Wang, 2006*); (*Baresi & Guinea, 2005*); (*van der Aalst, Pesic & Schonenberg, 2009*); (*Savickas & Vasilecas, 2018*); (*Milanovic, Gasevic & Rocha, 2011*); (*Bucchiarone et al., 2017*); (*Vasilecas, Kalibatiene & Lavbič, 2016*) |
| | 10.4. Does the paper present a clear description of an implementation with text, screen shots, and code lines? | (*Abderrahmane, Mili & Boubaker, 2014*); (*van der Aalst, Weske & Grünbauer, 2005*); (*Asuncion, Iacob & van Sinderen, 2010*); (*Awadid & Gnannouchi, 2015*); (*Ayora et al., 2016*); (*Ayora et al., 2012*); (*Boffoli et al., 2012*); (*Boffoli, Cimitile & Maggi, 2009*); (*Bögl, Natschläger & Geist, 2016*); (*Borrego & Barba, 2014*); (*Bucchiarone, Mezzina & Pistore, 2013*); (*Bucchiarone et al., 2011*); (*Cao et al., 2009*); (*Châtel, Malenfant & Truck, 2010*); (*Cherif, Djemaa & Amous, 2015*); (*Czepa et al., 2016*); (*Dadam & Reichert, 2009*); (*Deb, Chaki & Ghose, 2015*); (*Demeyer et al., 2010*); (*Gong & Janssen, 2012*); (*Haarmann et al., 2015*); (*Hallerbach, Bauer & Reichert, 2010*); (*Hermosillo, Seinturier & Duchien, 2010*); (*Jiang et al., 2016*); (*Kannengiesser et al., 2014*); (*Kapuruge, Han & Colman, 2011*); (*Khriss et al., 2008*); (*Kim, Kim & Kim, 2007*); (*Li & Du, 2016*); (*Liu et al., 2012*); (*Marconi et al., 2009*); (*Martinho, Domingos & Varajão, 2015*); (*Mattos et al., 2014*); (*Mejia Bernal et al., 2010*); (*Mounira & Mahmoud, 2010*); (*Natschläger et al., 2016*); (*Oberhauser, 2016*); (*Prasad et al., 2015*); (*Ramakrishnan, 2009*); (*Rong, Liu & Liang, 2008*); (*Santo Carvalho, Santoro & Revoredo, 2015*); (*Santo Carvalho et al., 2013*); (*Santos et al., 2013*); (*Santos et al., 2011*); (*Sarno et al., 2015*); (*Sprovieri et al., 2016*); (*Sun, Huang & Meng, 2011*); (*van Eijndhoven, Iacob & Ponisio, 2008*); (*Wang & Capretz, 2007*); (*Weidlich et al., 2011*); (*Xia & Wei, 2008*); (*Xiao et al., 2011*); (*Yoo et al., 2008*); (*Yousfi, Saidi & Dey, 2016*); (*Yuliang et al., 2009*); (*Gottschalk et al., 2008*); (*Maggi et al., 2012*); (*Pesic & van der Aalst, 2006*); (*Wang & Wang, 2006*); (*Baresi & Guinea, 2005*); (*van der Aalst, Pesic & Schonenberg, 2009*); (*Savickas & Vasilecas, 2018*); (*Milanovic, Gasevic & Rocha, 2011*); (*Bucchiarone et al., 2017*); (*Vasilecas, Kalibatiene & Lavbič, 2016*) |

(Continued)

| Table 15 (continued) | | |
| --- | --- | --- |
| RQ | Process dynamicity influencing features | References |
| | 10.5. Does the paper evaluate the results obtained? | (*Abderrahmane, Mili & Boubaker, 2014*); (*van der Aalst, Weske & Grünbauer, 2005*); (*Ayora et al., 2016*); (*Boffoli, Cimitile & Maggi, 2009*); (*Bucchiarone et al., 2011*); (*Czepa et al., 2016*); (*Deb, Chaki & Ghose, 2015*); (*Hallerbach, Bauer & Reichert, 2010*); (*Hermosillo, Seinturier & Duchien, 2010*); (*Jiang et al., 2016*); (*Mattos et al., 2014*); (*Mejia Bernal et al., 2010*); (*Santo Carvalho, Santoro & Revoredo, 2015*); (*Sprovieri et al., 2016*); (*Sun, Huang & Meng, 2011*); (*Ukor & Carpenter, 2009*); (*Yousfi, Saidi & Dey, 2016*); (*Bucchiarone et al., 2017*) |
| RQ4 RQ6 | 11. Discussion and conclusions: | |
| | 11.1. Does the paper present an appropriate discussion on dynamicity? | (*Abderrahmane, Mili & Boubaker, 2014*); (*Ayora et al., 2016*); (*Boffoli, Cimitile & Maggi, 2009*); (*Borrego & Barba, 2014*); (*Czepa et al., 2016*); (*Gong & Janssen, 2012*); (*Hermosillo, Seinturier & Duchien, 2010*); (*Mattos et al., 2014*); (*Xia & Wei, 2008*); (*Yousfi, Saidi & Dey, 2016*); (*Bucchiarone et al., 2017*); (*Vasilecas, Kalibatiene & Lavbič, 2016*) |
| | 11.2. Does the paper present grounded conclusions about dynamicity? | (*van der Aalst, Weske & Grünbauer, 2005*); (*Asuncion, Iacob & van Sinderen, 2010*); (*Awadid & Gnannouchi, 2015*); (*Ayora et al., 2012*); (*Boffoli et al., 2012*); (*Boffoli, Cimitile & Maggi, 2009*); (*Bögl, Natschläger & Geist, 2016*); (*Borrego & Barba, 2014*); (*Bucchiarone, Mezzina & Pistore, 2013*); (*Bucchiarone et al., 2011*); (*Cao et al., 2009*); (*Châtel, Malenfant & Truck, 2010*); (*Cherif, Djemaa & Amous, 2015*); (*Czepa et al., 2016*); (*Deb, Chaki & Ghose, 2015*); (*Demeyer et al., 2010*); (*Gong & Janssen, 2012*); (*Haarmann et al., 2015*); (*Hallerbach, Bauer & Reichert, 2010*); (*Hermosillo, Seinturier & Duchien, 2010*); (*Jiang et al., 2016*); (*Kannengiesser et al., 2014*); (*Kapuruge, Han & Colman, 2011*); (*Khriss et al., 2008*); (*Kim, Kim & Kim, 2007*); (*Li & Du, 2016*); (*Liu et al., 2012*); (*Marconi et al., 2009*); (*Martinho, Domingos & Varajão, 2015*); (*Mattos et al., 2014*); (*Mejia Bernal et al., 2010*); (*Mounira & Mahmoud, 2010*); (*Natschläger et al., 2016*); (*Oberhauser, 2016*); (*Prasad et al., 2015*); (*Ramakrishnan, 2009*); (*Rong, Liu & Liang, 2008*); (*Saidani & Nurcan, 2006*); (*Santo Carvalho, Santoro & Revoredo, 2015*); (*Santo Carvalho et al., 2013*); (*Santos et al., 2013*); (*Santos et al., 2011*); (*Sarno et al., 2015*); (*Sprovieri et al., 2016*); (*Sun, Huang & Meng, 2011*); (*Ukor & Carpenter, 2009*); (*van Eijndhoven, Iacob & Ponisio, 2008*); (*Wang & Capretz, 2007*); (*Weidlich et al., 2011*); (*Xia & Wei, 2008*); (*Xiao et al., 2011*); (*Yoo et al., 2008*); (*Yousfi, Saidi & Dey, 2016*); (*Yuliang et al., 2009*); (*Gottschalk et al., 2008*); (*Maggi et al., 2012*); (*Pesic & van der Aalst, 2006*); (*Wang & Wang, 2006*); (*Baresi & Guinea, 2005*); (*van der Aalst, Pesic & Schonenberg, 2009*); (*Savickas & Vasilecas, 2018*); (*Milanovic, Gasevic & Rocha, 2011*); (*Bucchiarone et al., 2017*); (*Vasilecas, Kalibatiene & Lavbič, 2016*) |

## ACKNOWLEDGEMENTS

The authors wish to thank Marlon Dumas for his feedback on a draft of this article.

### Funding

The authors received no funding for this work.

### Competing Interests

The authors declare that they have no competing interests.

### Author Contributions

- Diana Kalibatiene conceived and designed the experiments, performed the experiments, analyzed the data, performed the computation work, prepared figures and/or tables, authored or reviewed drafts of the paper, and approved the final draft.
- Olegas Vasilecas conceived and designed the experiments, analyzed the data, authored or reviewed drafts of the paper, and approved the final draft.

## Data Availability

This is a review article; there is no raw data.

## Supplemental Information

Supplemental information for this article can be found online at http://dx.doi.org/10.7717/peerj-cs.609#supplemental-information.

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
