# Peer review of "A survey on modeling dynamic business processes"

_PeerJ Computer Science, doi:10.7717/peerj-cs.609_

## Round 0.1 · original submission · Major Revisions

The reviews point to ways in which the manuscript falls short and can be improved further before it is ready for publication. Please do consider the reviews and revise the manuscript accordingly. Also, a separate document describing the changes made in the revision process is hugely welcome. This is also likely to shorten the review process in round 2.

·

Basic reporting

The manuscript investigated the level of dynamicity of dynamic business process modeling as well as the implementation in different existing approaches. The authors presented a literature review of approaches to modeling business process and provide a comparative evaluation of their level of dynamicity. Also, they identified the level of description and refinement in the analyzed dynamic business process approaches.
The manuscript is interesting. It is well prepared and presented.

Experimental design

Article content is within the scope of the journal.

Validity of the findings

The aims, the objectives and the outline of research are clearly defined. The literature is properly referenced and cited.

Additional comments

- Why the format of References’ citation differ? For example (Pesic & van der Aalst, 2006;Heinl et al., 1999) in Line 2020 and (2016b) in Line 203.

- Authors mention that ‘Due to technical limitations only free access sources were chosen’. Did the authors sent some requests to some authors of non-free access papers to obtain some interesting papers fruitful to the survey?

- Authors mention that one of the main results obtained during the survey of the selected papers consists of analysis of tools for DBP modeling and simulation. Why the authors did not go in depth for this point to deal with modeling languages and standards that cover the DBP?

- In Line 398, authors mention that they deal with surveys examining BP dynamicity from different perspectives. I am asking if these survey papers should be analysed different from non-survey papers? Or the same manner should be adopted?

- Authors mention briefly ‘process ontology’ but they do not compare proposed ontologies (richness and coverage). In the other hands, authors include context in the definition of DBP but do not enumerate how studied works deal with BP context (i.e, how they model and manage these contexts? Maybe by ontologies or other models?)

- Same remarks applies for ‘rules’

- Also in the definition of DBP, authors mention that ‘changes can be implemented with minimal delay’ Did the authors focus on this issue when analyzing and/or comparing existing works

Reviewer 2 ·

Basic reporting

- The structure of the article needs to be improved, many parts of the article are scattered.
- Previous work should be properly reviewed and the advantages of this article over them should be clearly stated

Experimental design

- The research question is well defined, but not well answered. It may not be clear to the readers what the authors are proposing, and how the solutions try to address the research question
- According to Table 6, 7 and 8, there is no source from recent years in the reviewed papers. Have sources been deleted by the authors? Or did they not exist? If they do not exist, it is suggested to revise the keywords.
- For previous works, advantages and disadvantages must be clear
- In the first part of the article, some views have been considered for the problem, but no specific classification has been done on their effective parameters. For example, implementation environments can be added to this category.

Validity of the findings

- The author should emphasize the advantages of this paper, more specifically, topic that introduced in this paper while not mentioned in other survey should be illustrated.
- Challenges and open issues must be clear.
- It is recommended that they be displayed as a percentage in the implementation and issues covered by the chart

Additional comments

he paper should be revised a lot and its structure and issues should be seriously re-examined

Reviewer 3 ·

Basic reporting

The review presented in this paper is limited to « open access » papers.

The study does not reference work which are very relevant for the study, example the contributions provided by Luca Sabatucci and Massimo Cossentino: Supporting Dynamic Workflows with Automatic Extraction of Goals from BPMN. ACM Trans. Auton. Adapt. Syst. 14(2): 7:1-7:38 (2019)

The pape is very long. While many tables and figures are put outside the paper, their content is an integral part of it. Many details could be removed from the paper, like the method of searching and selectiong papers, data extraction, etc. It would be better that the authors synthetise the content of the figures and the tables prensented outside of the paper.

Experimental design

The related work provide global positioning to existing work, but does not stress individual positioning to work very close to the provided review.

Validity of the findings

The conclusion does not identify future directions

Additional comments

The paper provides a very interesting review about modeling dynamic business process. The methodology is clear and well detailed and illustrated. Precise definitions and requirements about dynamic business processes have been given. Valuable conclusions have been also made. However, the paper does not consider interesting work very relevant to this study like those of Luca Sabatucci and Massimo Cossentino dealign with goal oriented dynamic modeling of BP.

In addition, the work fails to provide technical details about modeling of dynamic business processes. Merely statistical data about the number of paper dealing with different aspects of dynamic business processes.

---

## Round 0.2 · Minor Revisions

Please do a final check of the reviewers' comments and address them comprehensively. We are close to a final decision of 'accept', yet it is best to go thru a final round of checks & balances.

·

Basic reporting

All my remarks and suggestions have been well addressed by authors so that I recommend to accept the manuscript.

Experimental design

no comment

Validity of the findings

no comment

Reviewer 2 ·

Basic reporting

The changes requested by the reviewers have not been applied and most of the explanations have been added outside of the paper. the paper has not changed much

Experimental design

many the papers have been referred to in different places without a specific category, and it is suggested to consider a specific category for the articles that can be used by the reader

Validity of the findings

future directions has improved.

Additional comments

In my opinion, the opinions of the reviewers have not been applied correctly:
for reviewer 1:
Comment 1: In lines 94 to 106, the source of the Eshuis paper is addressed in several ways, one of which conforms to the standard. Please check the whole paper.
Comment 3: In some of the papers that have been reviewed, there has been a lot of scattering that suggests that the structure of the paper be controlled.
Reviewer 2:
Comment 1: It refers to the structure of the article and how to categorize the papers, many of them have been referred to in different places without a specific category, and it is suggested to consider a specific category for the articles that can be used by the reader.
Comment 4: Due to the length of the article process as well as the acceptance process and given that the publication of the article will take place in an optimistic state in 2021. It is better to update the resources so that the survey article can be used. Because now the article is for two years ago
Comment 5:Items that have been added like 353, 363 , 386, are not modified in the file, are not displayed in the changes, and are the same as before

---

## Round 0.3 · accepted · Accept

Thanks for your revision as per the reviews and submitting the materials in due time. I congratulate you on your work. Certainly we look forward to more submissions from you and your associates. All the best wishes...